



# Potential influence of overpressurized gas on the induced seismicity in the St. Gallen deep geothermal project (Switzerland)

Dominik Zbinden[1], Antonio Pio Rinaldi[1], Tobias Diehl[1], and Stefan Wiemer[1]

[1]Swiss Seismological Service, ETH Zurich, Switzerland

**Correspondence:** Dominik Zbinden (dominik.zbinden@sed.ethz.ch)

**Abstract.** In July 2013, the city of St. Gallen conducted a deep geothermal project that aimed to exploit energy for district heating and generating power. A few days after an injection test and two acid stimulations that caused only minor seismicity, a gas kick forced the operators to inject drilling mud to combat the kick. Subsequently, multiple earthquakes were induced on a fault several hundred meters away from the well, including a $M_L$ 3.5 event that was felt throughout the nearby population
centers. Given the occurrence of a gas kick and a felt seismic sequence with low total injected fluid volumes ($\sim$1200 m$^3$), the St. Gallen deep geothermal project represents a particularly interesting case study of induced seismicity. Here, we first present a conceptual model based on seismic, borehole and seismological data suggesting a hydraulic connection between the well and the fault. The overpressurized gas, which is assumed to be initially sealed by the fault, may have been released due to the stimulations before entering the well via the hydraulic connection. We test this hypothesis with a numerical model
calibrated against the borehole pressure of the injection test. We successfully reproduce the gas kick and the temporal and spatial characteristics of the main seismicity sequence that followed the well control operation. The results indicate that the gas may have destabilized the fault during and after the injection operations and could have enhanced the resulting seismicity. This study may have important implications for future deep hydrothermal projects conducted in similar geological conditions.

## 1 Introduction

Industrial injection and extraction projects causing anthropogenic earthquakes have increased during recent years and have been conducted closer to densely populated areas (Foulger et al., 2018). As a result, both the non-scientific and scientific community's interest in induced seismicity has risen dramatically. Anthropogenic earthquakes have been observed related to water impoundment, mining, geothermal power production, hydrocarbon extraction, hydraulic fracturing for shale gas extraction,
$CO_2$ sequestration, and wastewater injection (Ellsworth, 2013; Grigoli et al., 2017; Foulger et al., 2018). Industrial and societal problems arising from induced seismicity are twofold: On the one hand, large induced seismic events can be a risk to the population and cause damage to structures. For instance, in the United States mid-continent, several M>5 events have been recorded after wastewater injection, causing substantial damage to structures and harm to people (Yeck et al., 2017). On the





other hand, geo-energy projects may be jeopardized by lack of public support due to felt but only slightly damaging seismic
events. A striking example is the Enhanced Geothermal System (EGS) project in Basel, Switzerland, where seismicity was
induced immediately below the city, which led to the suspension of the entire project (e.g., Giardini, 2009). Recent history
clearly shows that the success of geo-energy, in particular geothermal projects, largely depends on the level at which we are
able to control induced seismicity (Kraft et al., 2009; Kwiatek et al., 2019). There is an urgent need to communicate transpar-
ently with the public and employ methods that will safely keep the seismicity to a tolerable level (Giardini, 2009; Lee et al.,
2019). Understanding the physics behind the induced seismicity is an important step necessary to assess the hazard and risk of
geo-energy projects and to develop methods to mitigate the seismicity. Hence, it is crucial to get a more accurate understanding
of the rock-fluid interaction occurring at reservoir depths.

To date, two main mechanisms are thought to be responsible for man-made earthquakes: (i) removing or adding mass
(e.g., mining, water impoundment), and (ii) injection or withdrawal of fluid (e.g., geothermal power, hydrocarbon extraction,
wastewater injection, $CO_2$ sequestration; Foulger et al. (2018)). For injection or extraction activities, pressure and temperature
changes influence the state of stress in the subsurface zones. This process is considered to be the main mechanism employed in
deep geothermal projects, where cold water injection and hot water or steam production can change the effective stress in and
around the reservoir and hence induce earthquakes. Several geothermal projects globally have induced seismicity (e.g., Baisch
et al., 2015; Evans et al., 2012; Grigoli et al., 2018; Jeanne et al., 2015); recently, a $M_w$ 5.5 earthquake was associated with
an EGS project struck the city of Pohang (South Korea) (Grigoli et al., 2018; Geological Society of Korea, 2019), the largest
earthquake recorded at an EGS site up to date (Kim et al., 2018). This earthquake has challenged recently proposed models that
relate the maximum expected seismic magnitude to the total injected fluid volume (Eaton and Igonin, 2018; Lee et al., 2019).
In contrast to the majority of geo-energy projects conducted so far, the main shock in Pohang lies well beyond the magnitude
threshold given by such models.
The deep geothermal project in St. Gallen conducted in 2013 shows some similarity with the Pohang event, given the
relatively strong induced seismicity (in terms of total released seismic moment) after the injection of rather small volumes of
fluid. A few days after an injection test and two acid stimulations that caused only minor seismicity, gas entered the well from
an unidentified source. The resulting gas kick was fought by pumping fresh water and heavier drilling mud into the well (e.g.,
Moeck et al., 2015). The well control injection induced multiple seismic events (Fig. 1), including a local magnitude $M_L$ 3.5
(moment magnitude $M_w$ 3.3; Diehl et al., 2014) earthquake that was distinctly felt throughout the population centers adjacent to
the well (e.g., Edwards et al., 2015). For St. Gallen, using McGarr's model (McGarr, 2014) that relates the maximum expected
magnitude $M_{max}$ to the product of the shear modulus $G$ (30 GPa following McGarr, 2014) and the total injected volume $V$
(ca. 1200 m$^3$ neglecting additional mud losses; Alber and Backers, 2015, and references therein), gives a $M_{max}$ of 3.0. Despite
the assumption of a relatively stiff fault, the main event ($M_w$ 3.3) is above the theoretically derived threshold. This calls for a
more thorough analysis of the hydro-mechanical processes taking into account multi-phase fluid flow to evaluate the potential
influence of the gas. Since we only have a limited knowledge of the deep subsurface (e.g., from boreholes and non-destructive
geophysical methods), numerical modeling can help shed light on otherwise hidden processes.





**Figure 1.** The St. Gallen induced seismicity sequence: (a) time series of the wellhead pressure in SG GT-1 and number of relocated events of the full catalog from July to the end of 2013. A: main seismicity sequence; B: more quiet post-injection period; C: seismic activity restarts due to cleaning and fishing operations. (b) Time series of the wellhead pressure in SG GT-1 and number of relocated events from 13 July to the end of July 2013. A.1: injection test; A.2: acid stimulations; A.3: gas kick and well control measures. (c, d, e) Time series of the injection rates and wellhead pressure in SG GT-1, as well as the number of relocated seismic events for the (c) injection test (14 July), (d) acid stimulations (17 July), and gas kick including the well control injection (19–20 July). The full relocated catalog is taken from Diehl et al. (2017).





In this study, we perform hydro-mechanical simulations to more accurately understand the causes of the induced seismicity during the St. Gallen deep geothermal project. Our aim is to propose and evaluate potential mechanisms that led to the seismicity and to reproduce the characteristics of the main sequence in July 2013 (denoted as phase A in Fig. 1a). First, we describe the temporal and spatial evolution of the seismic sequence associated with the injection. Secondly, we present a conceptual model for the induced seismicity in St. Gallen based on the earthquake catalog (Diehl et al., 2017), data from a 3-D seismic campaign (Heuberger et al., 2016), and data from the borehole St. Gallen (SG) GT-1 (Wolfgramm et al., 2015). We then present a 3-D numerical model using TOUGH2-seed (Rinaldi and Nespoli, 2017) that combines the multicomponent and multi-phase fluid flow simulator TOUGH2 (Pruess et al., 2012) with a geomechanical-stochastic model (Catalli et al., 2016; Gischig and Wiemer, 2013; Gischig et al., 2014; Goertz-Allmann and Wiemer, 2013). Our model is calibrated to the well pressure response during the injection test and the gas kick. We then simulate the injection test, the gas kick and the well control injection to reproduce the main seismic sequence at the end of July 2013. In our simulations, the primary focus is the potential effect of gas on the induced seismicity. Finally, we discuss our results in the context of future deep hydrothermal projects.

## 2 The deep geothermal project in St. Gallen

The St. Gallen deep geothermal project was conducted in the North Alpine Foreland Basin a few kilometers west of the city of St. Gallen (Fig. 2). The plan was to drill into the fractured damage zone within the St. Gallen Fault Zone (SFZ), a 20 km long fault system consisting of several steeply dipping normal and subsidiary apparent reverse faults striking NNE-SSW (Heuberger et al., 2016). The target formation was chosen to be the Malm carbonate layer (Upper Jurassic, top of the Mesozoic sediments at this site) located in the damage zone of two normal faults of the SFZ at a depth of about 4 km. The formation was expected to be sufficiently permeable to circulate water at rate of at least $50\,l\,s^{-1}$ (e.g., Hirschberg et al., 2015) per 200 m of drawdown in the well ($\sim$ 2 MPa; T. Bloch, pers. comm., 7 September 2019) without requiring permeability enhancement by EGS-type hydraulic stimulation. The choice of the reservoir was motivated by the fact that, at different sites in southeastern Germany, multiple hydrothermal systems are currently running successfully in similar geological conditions (Wolfgramm et al., 2015). A 3-D seismic survey carried out between 2009 and 2010 revealed that the SFZ tapers off in the crystalline basement, probably bounding a permo-carboniferous trough (PCT) below the Mesozoic sediments (Heuberger et al., 2016). The PCT is poorly defined in the seismic survey, since strong reflections from the sediments above prevented a detailed interpretation of the deeper horizons (Heuberger et al., 2016).

The St. Gallen region had experienced only minor natural seismicity prior to the project, with the largest being a $M_L$ 3.2 earthquake since 1984 (Diehl et al., 2017). Only two historic earthquakes with $M_w$>4.0 were reported in the vicinity of the geothermal site (Fäh et al., 2011). Nevertheless, in anticipation of a possible induced microseismic activity, the regional seismic network was densified locally at the beginning of 2012 with a short period borehole sensor and five broadband surface stations within a radius of 12 km around the geothermal well. During the stimulation phase in July 2013, the network was further extended by seven short period surface stations (Diehl et al., 2017; Edwards et al., 2015).





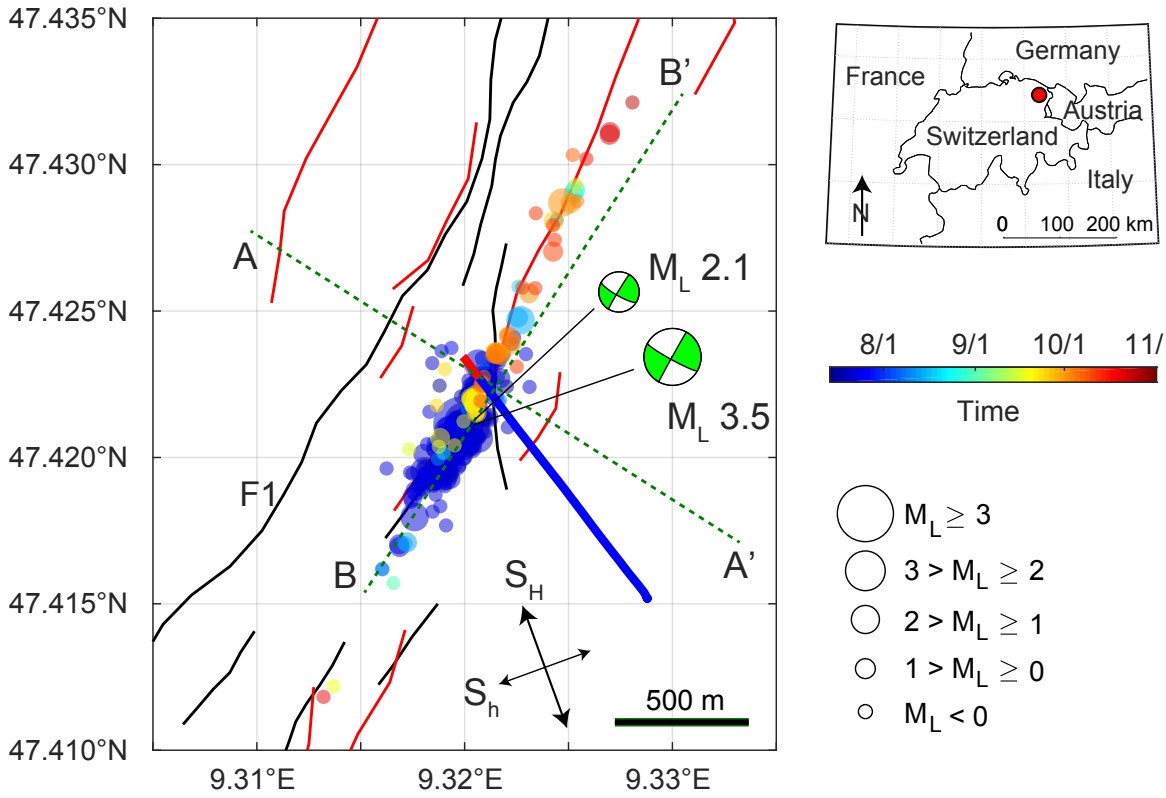

**Figure 2.** Map view of the relocated events color-coded by origin time (format month/day). The fault plane solutions of the $M_L$ 3.5 and $M_L$ 2.1 events (Diehl et al., 2017) are denoted by the green focal mechanisms. Red (top-Malm) and black (bottom-Muschelkalk) lines illustrate the faults of the SFZ including a large fault (F1) bounding the permo-carboniferous trough (Heuberger et al., 2016). The cased and open section of borehole GT-1 are denoted by the thick blue and red lines, respectively. The black arrows sketch the orientation of the minimum and maximum principal stress after Moeck et al. (2015). The red dot in the European map (top right) marks the location of the St. Gallen deep geothermal project.

In the beginning of July 2013, the drilling of the SG GT-1 well was completed to a true vertical depth (TVD) of about 4.2 km without inducing any seismic events (depth is defined relative to the top of the borehole throughout the paper). The open section extended from 3.8 to 4.2 km TVD within the Malm formation. In order to get an estimate of the hydraulic properties of the reservoir, an injection test was performed on 14 July (phase A.1 in Fig. 1b and Fig. 1c). Water was injected with a step-wise increasing rate reaching a maximum of $53 \, l \, s^{-1}$. A total of 175 m$^3$ of water was pumped into the subsurface leading to a pressure
increase from 36.7 MPa to 46.5 MPa ($\Delta P = 9.8$ MPa) and inducing some microearthquakes with six being precisely located (Diehl et al., 2017). These were the first seismic events recorded since microseismic monitoring began in 2012. On 17 July, two acid stimulations were performed in sections where permeable fractures were expected, and a total of 150 m$^3$ hydrochloric

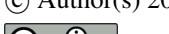



acid and 140 m$^3$ of water was injected (phase A.2 in Fig. 1b and Fig. 1d). The first acid stimulation was conducted in the lower
Malm at a depth of 4.15 km, reaching maximum injection rates of 42 l s$^{-1}$ and a wellhead pressure increase of 8.8 MPa. The

second acid stimulation was performed in the upper Malm with injection rates up to 24 l s$^{-1}$ and a pressure increase up to 6.3
MPa. During the first acid stimulation, only a few microearthquakes with $M_L$<0.0 were recorded, two of them being relocated
at a depth of approx. 4.9 km. The second acid stimulation led to much more seismicity: 19 microearthquakes with $M_L$<0.5 were
relocated until the morning of 19 July at an average depth of ca. 4.6 km (Diehl et al., 2017), i.e., shallower than the microevents
during the first acid stimulation. The number and magnitude of the seismic events was well within the range expected for the

injected fluid volumes. On 19 July at noon, gas (roughly 90 % methane; Wolfgramm et al., 2015) entered the well from an
unknown source; for security reasons, the well was closed immediately. Since the wellhead pressure increased to between 8 and
9 MPa due to the inflowing gas, fresh water and heavier fluids were pumped into the well at rates of up to 20 l s$^{-1}$ over 15 hours
(total volume of approx. 700 m$^3$) to prevent further gas inflow (phase A.3 in Fig. 1b and Fig. 1e). This injection successfully
decreased the wellhead pressure. However, numerous seismic events were induced in the evening of 19 July, one triggering

110  the yellow threshold of the installed traffic light system (TLS), meaning that injection should be stopped immediately (e.g.,
Diehl et al., 2017; Obermann et al., 2015). Nevertheless, injection was continued until the early morning of 20 July to keep the
wellhead pressure at about 3 MPa; stopping the injection would probably have led to an increase in the wellhead pressure due
to further gas inflow. Seismicity continued and a $M_L$ 2.1 event was induced at 2.40 am on 20 July, followed by multiple smaller
events. At 5.30 am, after more water, including heavy components, was injected, a $M_L$ 3.5 event was induced. Subsequently,

115  the pressure normalized and the injection was stopped. The main shock was followed by almost 200 seismic events (relocated)
until the end of July (Diehl et al., 2017) (Fig. 1b). After a preliminary analysis of the incidents, the geothermal project was
temporally suspended. The main seismic sequence was followed by only minor seismicity between August and mid-September
2013 (phase B in Fig. 1a). Cleaning and fishing operations in the borehole from mid-September to mid-October likely led to a
temporal increase of the seismicity rate due to some mud losses (phase C in Fig. 1a). During a production test conducted in the

end of October, the seismicity ceased completely. Thereafter, the operators decided to permanently suspend the project because
the permeability of the reservoir was much too low (flow rate $< 6$ l s$^{-1}$ for a drawdown of approx. 1500 m in the borehole ($\sim 15$
MPa); Wolfgramm et al., 2015) and the gas content was found to be too high to maintain flow rates sufficient for economic
feasibility of a hydrothermal power plant. EGS-type hydraulic stimulation (i.e., hydro-shearing) was not considered because
the risk of further seismicity was judged to be too high (e.g., Moeck et al., 2015).

The spatial distribution and evolution over time of the induced seismicity is illustrated in Fig. 2. The seismic sequence
extends along a relatively narrow band (width of several hundreds of meters) striking SSW-NNE and dipping approximately 70°
to WNW. During the injections in July 2013, the seismicity mainly propagated in the south-west direction from the borehole.
Only in August to October 2013 the seismicity started to propagate to the north-east, but at much smaller propagation velocities
(Diehl et al., 2017). The main focal mechanism in St. Gallen is strike-slip, which can be deduced from the two largest events
of the sequence (Fig. 2). To measure the local stress field, Moeck et al. (2015) performed in situ stress estimations for borehole

SG GT-1 and determined that the maximum ($S_H$) and minimum ($S_h$) principal stresses were horizontal trending $160 \pm 12°$
and $70 \pm 12°$, respectively. The vertical stress ($S_v$) was estimated to be intermediate with a magnitude of 98 MPa at a depth





of ca. 3.9 km. Furthermore, the mean stress ratios of $S_H = 1.41 \cdot S_v$ and $S_h = 0.61 \cdot S_v$ were calculated at 4.1 km TVD. On a more regional scale, Kastrup et al. (2004) inverted focal mechanisms of naturally occurring earthquakes over the last 50 years

to obtain the orientation of the three principal stresses in the Northern Alpine Foreland. They obtained a trend of about $160°$ to $170°$ for the maximum horizontal stress – in good agreement with the in situ stress estimations. Assuming a straight planar fault, we calculate the best fit strike orientation (least squares regression) to be approximately $210°$ (Fig. 2), coinciding with the focal mechanisms of the two largest induced events. Hence, in comparison to the local stress field, $S_H$ intersects the fault with an angle of about $50°$.

**3   Conceptual model**

First, we describe the conceptual model of the stimulation phase, the gas kick and the subsequent well control injection in July 2013. Clearly, there is a temporal correlation between the seismicity and the fluid injected during the injection test, the acid stimulations and the well control measures (Fig. 1). With regard to the spatial distribution, despite the estimated vertical and horizontal absolute location uncertainties of 0.15 km and 0.1 km, respectively (Diehl et al., 2017), most of the seismicity

occurred in the pre-Mesozoic basement below a depth of 4.4 km (Fig. 3). Hence, considering the mean locations, the seismicity is separated from the well bottom (at a depth of ca. 4.2 km) by a minimum distance of about 0.3 km. The main shock ($M_L$ 3.5) was located at a depth of about 4.6 km, 0.2 km further south-west with respect to the top of the open well section. The spatial gap between the borehole and the seismicity suggests that the seismic events were either triggered remotely by poroelastic stress changes or by a hydraulic connection (e.g., a fracture zone or a damage zone of a fault; Zbinden et al., 2019). Temperature

and gamma-ray anomalies from borehole logs support a connection to the lower Mesozoic sediments and possibly to the pre-Mesozoic basement (Wolfgramm et al., 2015). The temperature anomalies indicate the presence of major inflow zones intersecting with the borehole, the most prominent one at the upper part of the Malm (at a depth of 3.9 km). In the gamma-ray log, thorium anomalies suggest a connection between the borehole and the Dogger layer (underlying the Malm reservoir) or deeper (Wolfgramm et al., 2015). From a geometrical analysis, Diehl et al. (2017) suggested that at least one fault mapped

by the seismic survey (Heuberger et al., 2016) might act as a hydraulic connection. The presence of a hydraulic connection would suggest that the injected fluid moved from the borehole toward the fault, resulting in an increase in pore pressure and a decrease in effective normal stress, eventually destabilizing the fault and leading to seismicity (Zbinden et al., 2019). Models have shown that the seismicity can be significantly stronger for hydraulically connected faults than for unconnected faults (Chang and Segall, 2016). Since the seismic response in St. Gallen was unexpectedly intense, it would support the hypothesis

of a hydraulic connection.

With respect to the gas kick, the gas may have originated from the PCT located below the Mesozoic sediments, which contains high concentrations of organic carbon (Heuberger et al., 2016; Wolfgramm et al., 2015). The gas might then have migrated upwards over geologic time, eventually reaching an impermeable seal preventing the gas from moving higher up. The stratigraphy in the St. Gallen region indicates that the lower Mesozoic sediments (Keuper, Lias and Dogger) could serve

as a caprock to the gas reservoir. Additionally, the fault may have acted as a lateral seal to prevent the gas from reaching the



**Figure 3.** Conceptual model of the stimulation phase, the gas kick and the well control measures in July 2013. (a, b) Seismicity of the injection test (14 July) on a profile (a) normal to (A–A' in Fig. 2) and (b) along (B–B' in Fig. 2) the fault. The blue shaded area denotes the region affected by the direct pressurization, while the red shaded area depicts the possible location of the gas as described in the text (shown in all subfigures). The question mark next to the deeper events indicates that they cannot be fully explained by a single hydraulic connection. (c, d) Seismicity due to the acid stimulations (17 July) on a profile (c) normal to and (d) along the fault. The yellow dots correspond to the seismicity induced during the first acid stimulation, while the cyan dots mark the events induced during the second acid stimulation. The red shaded area in (d) is the region affected by the proposed seal breach through which the gas may have migrated. (e, f) Seismicity during and after the well control measures (19–31 July) on a profile (e) normal to and (f) along the fault. The injected fluid and the gas that is intruding the fault could have caused the seismic sequence including the $M_L$ 3.5 main shock. Depth is defined relative to the top of the borehole (i.e., the free surface).





hydraulic connection prior to the stimulations (Fig. 3). The gas kick first occurred two days after the acid stimulations and five days after the injection test, whereas no gas was observed during the drilling of the well (Naef, 2015). This is an indication that a hydraulic connection was indeed present, enabling the gas to enter the borehole from a greater depth. On the other hand, it implies that the hydraulic connection was only established after the injection test and the acid stimulations. Possibly,

the stimulations enhanced the permeability of the hydraulic connection; the resulting seismicity as well as the acidification breached the fault seal, opening up a pathway for the gas to reach the well. This implies that the gas took the same path as the injected fluid, but in the opposite direction (Fig. 3).

In summary, we propose that the following processes could have led to the main seismicity sequence (14–31 July 2013) in St. Gallen.

1. During the injection test and the acid stimulations, the fault was pressurized by a highly permeable hydraulic connection (whose permeability is enhanced by the pressurization and the acid treatment).

2. The pressure increase on the fault led to a decrease in the effective normal stress and caused minor seismicity (Fig. 3a–Fig. 3d).

3. Due to the acidification and shear slip associated with the fault reactivation, a pathway opened up for the gas to reach the
borehole through the hydraulic connection and caused the gas kick (Fig. 3c and Fig. 3d).

4. The fluid injected into the well during the control measures and the gas destabilized a larger patch on the fault leading to the seismic sequence that included the $M_L$ 3.5 event (Fig. 3e and Fig. 3f).

5. The injection of water and drilling mud during the well control measures probably clogged the hydraulic connection and stopped the gas kick.

In our conceptual model, we show one distinct fracture zone that intersects with the borehole at the upper Malm where a strong temperature anomaly was observed. However, the logs show that other permeable structures might exist at the depth of the lower Malm. Moreover, for both the injection test and the first acid stimulation, Diehl et al. (2017) observed that the seismicity was initiated at a greater depth (between 4.8 and 5.0 km) compared to later events. The first acid stimulation was performed in the lower section of the Malm and induced two microearthquakes at a depth of about 4.9 km (Fig. 3c and Fig. 3d). The

second acid stimulation was performed in the upper Malm section, inducing seismic events above a depth of 4.7 km. For the injection test and the well control measures, where water and drilling mud was injected into the cased section and the well was pressurized equally, seismic event locations ranged from a depth of approximately 4.4 to 4.9 km. Given these observations, the presence of a second permeable connection seems possible: the first structure connects the upper Malm at the borehole with the reactivated fault at a depth of about 4.5 km, while a second permeable structure may connect the lower Malm with the fault

at a depth of about 4.8 km. The second hydraulic connection can explain the fast seismic response to the stimulations in the lower part of the fault, which cannot be explained by only one hydraulic connection. Despite these observations, Diehl et al. (2017) proposed that the vertical offset of this cluster is a location artifact, which can be explained by the presence of a local





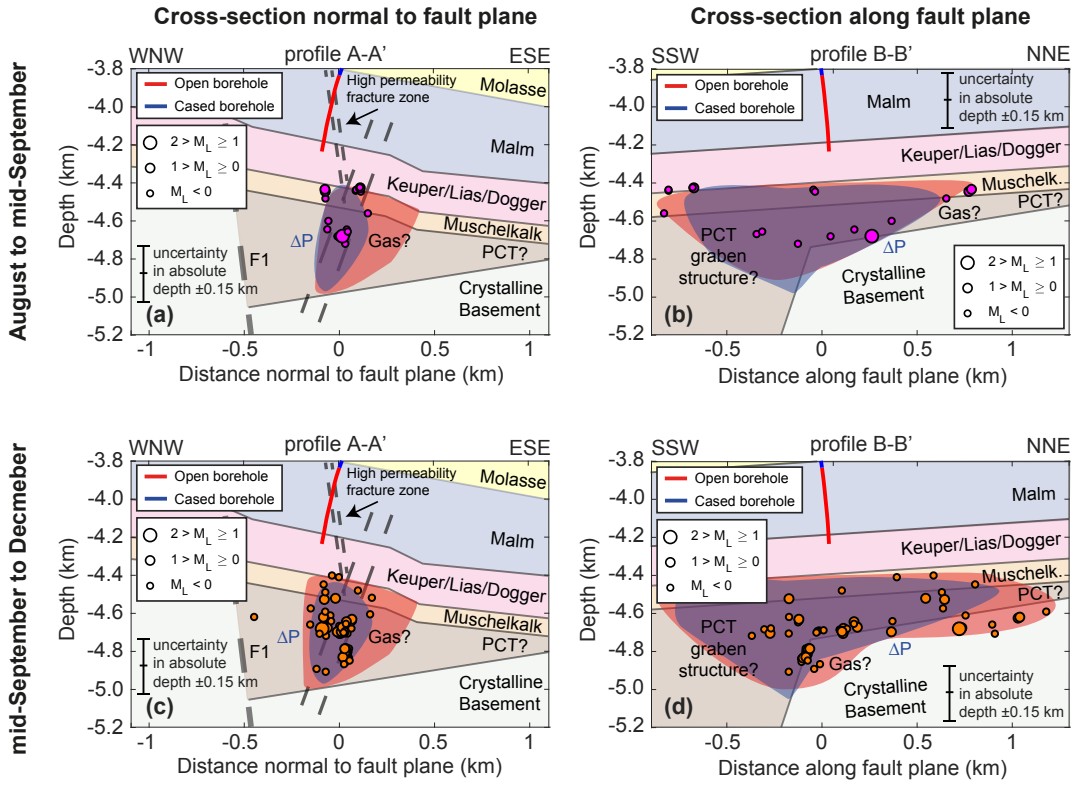

**Figure 4.** Conceptual model of the post-stimulation phase from August to December 2013. (a, b) Seismicity for the period 1 August to 15 September on a profile (a) normal to (A–A' in Fig. 2) and (b) along (B–B' in Fig. 2) the fault. The blue shaded area denotes the region affected by the direct pressurization, while the red shaded area depicts the possible location of the gas (shown in all subfigures). The seismicity in July is mainly located on the southwestern side of the borehole, whereas the seismic events after 1 August start to propagate to the northeast. (c, d) Seismicity of the period from 16 September to the end of 2013 on a profile (c) normal to and (d) along the fault. Borehole operations associated with some mud loss may be responsible for the increase of seismicity in September and October. The seismicity propagates further to the northeast, but ceases during a production test at the end of October. Depth is defined relative to the top of the borehole (i.e., the free surface).

$v_p/v_s$ velocity anomaly. For this reason, it remains unclear whether a second hydraulic connection was present. For the sake of simplicity, we choose to perform the numerical simulations of the gas kick and well control measures with only a single

hydraulic connection, whereas the injection test is simulated with an additional model containing two connections as a test case (see below).

Figure 4 shows the conceptual model for the post-stimulation phase from August to December 2013. The seismicity of the main sequence (July 2013) was mainly distributed south-west of the injection well, although some minor events induced between August and mid-September 2013 were located north-east of it (Fig. 4a and Fig. 4b). In order to explain this observation,





Diehl et al. (2017) argued that a seal, represented by a mapped fault intersecting with the reactivated fault in the area immediately beneath the borehole (Heuberger et al., 2016), may have been breached by the 20 July main shock. Only after this event could the fluid flow to the north-east. Alternatively, it can be argued that the permeability of the reactivated fault was lower to the northeast, since the PCT probably tapers off in this direction (Fig. 4b and Fig. 4d, see also Fig. 11 in Heuberger et al., 2016), and hence the fault may contain lower permeability components from the suggested crystalline horst. In this case, the pressure

front would then propagate slower to the northeast, consistent with the seismic observations. From mid-September to the end of October 2013, the seismicity increased again in the area below the borehole, although not drastically (Fig. 4c and Fig. 4d). At the same time, mud loss was observed in borehole SG GT-1 due to some fishing and cleaning operations. This indicates that the hydraulic connection, possibly clogged by the well control injection, may have been reopened and the mud was able to reach the fault plane. Subsequently, the seismicity continued to propagate further to the northeast, while no seismic activity was

observed southwest of the borehole. This may be associated with renewed gas movement due to the pressure changes caused by the mud. It is possible that the mud clogged potential pathways to the southwest, and hence the gas could only propagate to the northeast, inducing the seismicity in this region. Alternatively, the mud itself and some gas may have propagated to both the southwest and northeast, but only induced seismic events in the northeast because the fault might possibly terminate in the southwest. We do not have data that supports any of these hypotheses and, therefore, we cannot assess whether the seismicity

in the post-injection period was induced by the gas, the mud or a mixture of both. The seismicity in St. Gallen ceased during a production test. It was observed that the entire seismic sequence was constrained below a depth of about 4.4 km (Fig. 3 and Fig. 4), which coincides with the top boundary of the Muschelkalk. The reason for this may be (i) a stiffness contrast of the softer Keuper, Lias and Dogger layers with respect to the stiffer Muschelkalk and Malm layers, which absorb most of the tectonic stress, leading to a lower differential stress in the softer layers and therefore to a less critically stressed fault in this

region (Hergert et al., 2015), or (ii) a lower permeability of the reactivated fault due to the surrounding low-permeability layers (Keuper, Lias and Dogger) and thus a smaller pressurization above the Muschelkalk – as assumed in our conceptual model.

## 4    Model setup and calibration

The numerical simulations are performed using TOUGH2-seed (Rinaldi and Nespoli, 2017). The seismic module of TOUGH2-seed is switched off during the simulation of the injection test and the gas kick to simplify the calibration of the model; however,

we use the full model during the simulation of the main seismic sequence.

### 4.1    TOUGH2-seed

TOUGH2-seed couples TOUGH2 with a stochastic-geomechanical model (Catalli et al., 2016; Gischig and Wiemer, 2013; Gischig et al., 2014; Goertz-Allmann and Wiemer, 2013). TOUGH2 is a multi-phase, multicomponent, and heat transport numerical simulator based on a first-order implicit finite difference scheme in time and an integral finite difference method in

space (Pruess et al., 2012). After the computation of the pressure field in TOUGH2, the solution is passed on to the seed model for each time step. TOUGH2-seed incorporates two different modules: (i) the hydraulic module that is similar to TOUGH2



but accounts for pressure-dependent permeability, and (ii) a seismic module that simulates induced seismicity and can be switched off if no seeds are distributed in the model. *Seeds* are uniformly randomly distributed potential failure points with prescribed stress and failure conditions. The hydraulic and seismic modules can be fully coupled if the permeability is chosen

to be dependent on the seismicity (Gischig et al., 2014; Rinaldi and Nespoli, 2017). TOUGH2-seed has been successfully applied to the EGS project in Basel, where most of the characteristic behavior of the induced seismicity could be reproduced (Rinaldi and Nespoli, 2017). The main advantage of using a stochastic model is that uncertainties can be assigned to different model parameters (e.g., stress magnitude, fault orientation, friction) depending on how well they are constrained by field data. Synthetic earthquake catalogs can be obtained after each simulation, allowing a comparison to observed earthquake data.

In this study, the pressure $P$ strictly refers to the average pore pressure (summation over fluid phases) exerted by the gas and liquid phases (e.g., Kim et al., 2013), written as

$$P = S_g \cdot P_g + (1 - S_g) \cdot P_w \tag{1}$$

where $S_g$ is the gas saturation, $P_g$ the gas pressure and $P_w$ the water pressure. Here we ignore the interfacial energy caused by the two-phase system (Kim et al., 2013, and references therein). For the hydraulic module, we account for pressure-dependent

permeability following Rinaldi and Nespoli (2017):

$$\kappa_{hm} = \kappa \cdot \exp\left[C_1\left(\frac{\phi_{hm}}{\phi_0} - 1\right)\right] \tag{2}$$

$$\phi_{hm} = (\phi - \phi_r) \cdot \exp(\alpha \Delta P) + \phi_r \tag{3}$$

where $C_1$ and $\alpha$ (Pa$^{-1}$) are scaling parameters. The updated permeability $\kappa_{hm}$ is a function of the initial permeability $\kappa$ and

the ratio of the actual porosity $\phi_{hm}$ and the initial porosity $\phi$. $\phi_{hm}$ is exponentially dependent on the pore pressure change $\Delta P$ and linearly dependent on the difference between $\phi$ and the residual porosity $\phi_r$. Note that the permeability and porosity changes in Eq. (2) and Eq. (3) are reversible (i.e., purely elastic). For the seismic module, TOUGH2-seed takes into account the full 3-D stress state formulation as well as the static Coulomb stress transfer (Catalli et al., 2016; Rinaldi and Nespoli, 2017). TOUGH2 solves for the pore pressure in each model element and transfers it to the seed model, where the new stress state of

the seeds is computed according to the analysis of effective stress (Terzaghi, 1923):

$$\sigma'_{ij} = \sigma_{ij} - P \tag{4}$$

where $\sigma'_{ij}$ is the effective stress tensor and $\sigma_{ij}$ is the total stress tensor. The current version of TOUGH2-seed does not account for poroelasticity, i.e., the total stresses do not change due to pressure and vice versa. The initial stress state on the seeds follows a normal distribution around a mean regional stress field in order to account for stress heterogeneity that can be present

in reality. The seeds are distributed on a fault and their orientation is normally distributed around the strike ($\gamma$) and dip ($\theta$) of the fault. From the orientation and the given effective stress tensor of the seeds, the effective normal stress $\sigma'_n$ and the shear



stress $\tau$ can be computed (e.g., Zoback, 2010). A seed is triggered if a Mohr-Coulomb failure criterion is reached, i.e., when the shear stress exceeds the shear strength $\tau_s$ (critical shear stress):

$$\tau_s = c + \mu_s \cdot \sigma'_n \tag{5}$$

where $c$ is the cohesion and $\mu_s$ the static friction coefficient, which follows a normal distribution (i.e., the strength of the seeds varies). Here, we assume a coefficient of friction of $0.6 \pm 0.05$ and a cohesion of 1 MPa. Moreover, following Gischig and Wiemer (2013), we define a criticality threshold $\mu_c$, so that the stress states of the seeds at the beginning of the simulation have a certain gap to the failure criterion. A moment magnitude is assigned to each triggered seed, randomly chosen from a Gutenberg-Richter (GR) distribution with a fixed $b$-value of 1.2 and a magnitude of completeness ($M_c$) of 0.8 obtained from
the St. Gallen catalog and recalculated for $M_w$ using a corrected maximum likelihood method (e.g., Marzocchi and Sandri, 2003, and references therein). A shear stress drop $\Delta\tau$ is calculated for each reactivation given by Gischig et al. (2014):

$$\Delta\tau = \Delta\tau_{coeff} \cdot \left( \frac{\tau - c}{\mu_s} \right) \tag{6}$$

where $\Delta\tau_{coeff}$ is a coefficient determining how much shear stress is released. A new stress state is then computed on the reactivated seed and on all neighboring seeds that are affected by the static stress transfer. After the evaluation of the triggered
seeds, the permeability is updated and TOUGH2 solves for the pressure distribution of a new time step before the coupling is repeated.

## 4.2    Numerical model setup

The 3-D hydro-geological model consists of a fracture zone connecting the upper part of the open section of the borehole (at a depth of approx. 3.9 km) with a fault plane intersecting a caprock layer (Fig. 5). The fracture zone dips 77° and is therefore
a conjugate plane of the reactivated fault, which itself dips 70° toward the WNW. The fracture zone extends over a length of about 920 m along the dip from a depth of 3.6 km to 4.5 km, where it is bound by the fault. The top corresponds to the base of the Molasse sediments (not modeled), where the fracture zone is presumed to taper off. Furthermore, the fracture zone is assumed to have an along-strike length of 250 m. The fracture zone is modeled as a high permeability, low porosity and high compressibility equivalent porous medium with a thickness of 20 m. The permeability in the fracture zone is pressure-
dependent to account for fracture opening during injection (initial permeability, pore compressibility and $\alpha$ in Eq. (3) are calibration parameters). We assume a very small initial porosity of $3 \cdot 10^{-5}$, corresponding to a total void space of 600 microns within the 20 m thick equivalent porous medium domain (e.g., 10 fractures with an aperture of 60 microns each). The fault zone consists of a two-sided, $2 \cdot 20$ m wide permeable damage zone ($\kappa = 10^{-14}$ m$^2$) and a 5 m wide, impermeable fault core ($\kappa = 10^{-22}$ m$^2$), and it extends over the entire width and length of the model (2 km in the Y-axis and ca. 1.9 km in the Z-
direction). The fault intersects a slightly inclined, very low permeability caprock ($\kappa = 10^{-22}$ m$^2$) corresponding to the lower Mesozoic sediments (Keuper, Lias and Dogger). A host rock layer representing the Malm reservoir surrounds the open section of the well. For simplicity, we assume that the domain below the caprock has the same properties as the host rock (Fig. 5a). The well is approximated as a high porosity and high permeability porous medium, under the assumption of negligible dynamic



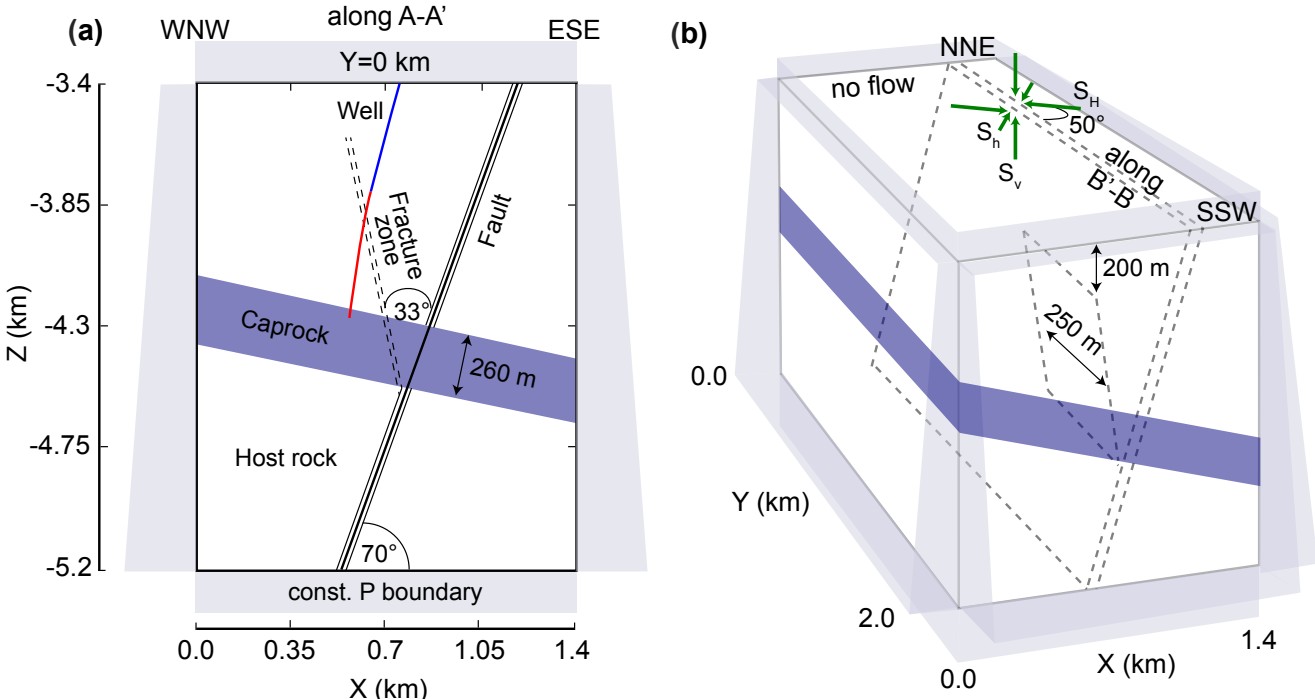

**Figure 5.** Schematic of the 3-D numerical model: (a) Slice at Y=0 km (along profile A–A' in Fig. 2). The well is indicated by the blue (cased section) and red lines (open section). (b) Entire 3-D model along the strike of the fracture zone and fault (along profile B–B' in Fig. 2). The dark green arrows indicate the orientation of the mean stress field used for the TOUGH2-seed simulations.

pressures within the wellbore (well permeability is a calibration parameter; Rinaldi et al., 2017). For the multi-phase flow

modeling, we use the relative permeability curves of Corey (1954) and capillary pressure functions of van Genuchten (1980). A summary of the hydraulic parameters is given in Table 1.

In accordance with the in situ conditions measured in borehole SG GT-1 in St. Gallen (Wolfgramm et al., 2015), we initialize the pressure at approx. 37 MPa and the temperature at 143 °C at the top of the open well section. The temperature linearly increases with depth by 35.5 °C km$^{-1}$ and is held constant during the injection. Although injection of low temperature fluid

can cool the rock and influence both the stress and pressure conditions (e.g., Ghassemi and Tao, 2016; Hopp et al., 2019; Rinaldi et al., 2015), we do not expect a significant cooling effect over the relatively short-term and small volume injection that occurred in St. Gallen. We choose the boundaries to be open for fluid flow everywhere except at the boundary Y=0 km (symmetry boundary), where we apply no flow conditions (Fig. 5b). In order to model the multi-phase fluid system, we employ an equation of state with water and air. We create an initial steady-state condition of a gas plume below the caprock on the

right side of the fault by simulating gas inflow with a pressure of 70 MPa at the lower boundary into the right damage zone for one million years (0.1 km long line source along the Y-axis). To prevent gas leakage, we assign a high capillary entry pressure to the fault core and the caprock. Figure 6 shows the initial pressure distribution and gas saturation at three different vertical





**Table 1.** Predefined and calibrated hydraulic parameters of the numerical model

|  | Host rock | Fracture zone | Damage zone | Fault core | Caprock | Well open/cased |
|---|---|---|---|---|---|---|
| Initial porosity $\phi$ (-) | 0.05 [a] | $3 \cdot 10^{-5}$ | 0.10 | 0.01 | 0.01 | 0.90/0.99 |
| Residual porosity $\phi_r$ (-) | 0.05 | 0.00 | 0.10 | 0.01 | 0.01 | 0.90/0.99 |
| Initial permeability $\kappa$ (m$^2$) | $10^{-18}$ [b] | $1.3 \cdot 10^{-14}$ [c] | $10^{-14}$ | $10^{-22}$ | $10^{-22}$ | $1.5 \cdot 10^{-8}$ [c]/$10^{-5}$ |
| $C_1$ (-) | – | 57.3 [c] | – | – | – | – |
| $\alpha$ (Pa$^{-1}$) | – | $10^{-8}$ | – | – | – | – |
| Pore compressibility $c_\phi$ (Pa$^{-1}$) | $5 \cdot 10^{-10}$ | $3.7 \cdot 10^{-8}$ [c] | $5 \cdot 10^{-10}$ | $5 \cdot 10^{-9}$ | $5 \cdot 10^{-9}$ | $5 \cdot 10^{-11}$ |
| Residual gas saturation $S_{gr}$ (-) | 0.05 | 0.05 | 0.05 | 0.05 | 0.05 | 0.05 |
| Residual liquid saturation $S_{lr}$ (-) | 0.3 | 0.05 | 0.1 | 0.3 | 0.3 | 0.05 |
| van Genuchten (1980), $P_0$ (MPa) | 2.0 | $2.4 \cdot 10^{-3}$ | 0.02 | 19.9 | 19.9 | $10^{-6}$ |
| van Genuchten (1980), $m$ (-) | 0.457 | 0.457 | 0.457 | 0.457 | 0.457 | 0.457 |

[a] Moeck et al. (2015); [b] Wolfgramm et al. (2015); [c] calibrated

cross-sections. The pressure increase is approximately linear with depth (approx. 9.1 MPa km$^{-1}$) above the caprock, while it is highly disturbed below the caprock by the *overpressurized* gas plume (i.e., pressurized with respect to an undisturbed state without gas). At the fracture zone/fault intersection at Y=0 km (at a depth of 4.5 km), the pressure on the right side of the fault exceeds the pressure on the left by about 7 MPa. This pressure difference becomes less pronounced toward the boundary at Y=2 km (Fig. 6a to Fig. 6c). The plume is completely gas saturated in the right damage zone and in the adjacent host rock at Y=0 km, while the size of the plume decreases along the strike of the fault (Fig. 6d to Fig. 6f). The domain below the caprock on the left side of the fault has an initial gas saturation of 5 %.

The stress input for the seed model is based on the findings of Moeck et al. (2015). The maximum principal stress $S_1$ is horizontal and has a mean trend of 160°, the intermediate principal stress $S_2$ is equal to the vertical stress $S_v$ and the minimum principal stress $S_3$ is horizontal with a trend of 70°. The seeds are randomly distributed on the lower part of the fault ($Z \leq -4.5$ km and $Y \leq 1.0$ km) and in the immediate surroundings, not more than 90 m normal distance to the fault core. Due to uncertainties in the exact orientation of the principal stresses with respect to the reactivated fault plane, we assign a standard deviation of 12° on both the dip (mean of 70°) and the strike of the seeds (the latter is equivalent to a 12° uncertainty on the horizontal stress orientations). The mean strike of the seeds is 210°, thus exhibiting an average angle of 50° with respect to $S_1$. For the vertical stress magnitude, starting from $S_v = 85.3$ MPa at a depth of 3.4 km, we use a stress gradient of 26.0 MPa km$^{-1}$ according to density estimations from borehole samples (rock density $\rho = 2650$ kg m$^{-3}$; Alber and Backers, 2015, and references therein). We set $S_H = (1.41 \pm 0.39) \cdot S_v$ and $S_h = (0.61 \pm 0.08) \cdot S_v$, while the standard deviation of $S_v$ is set to 0.03. A list of the seed model parameters is given in Table 2.





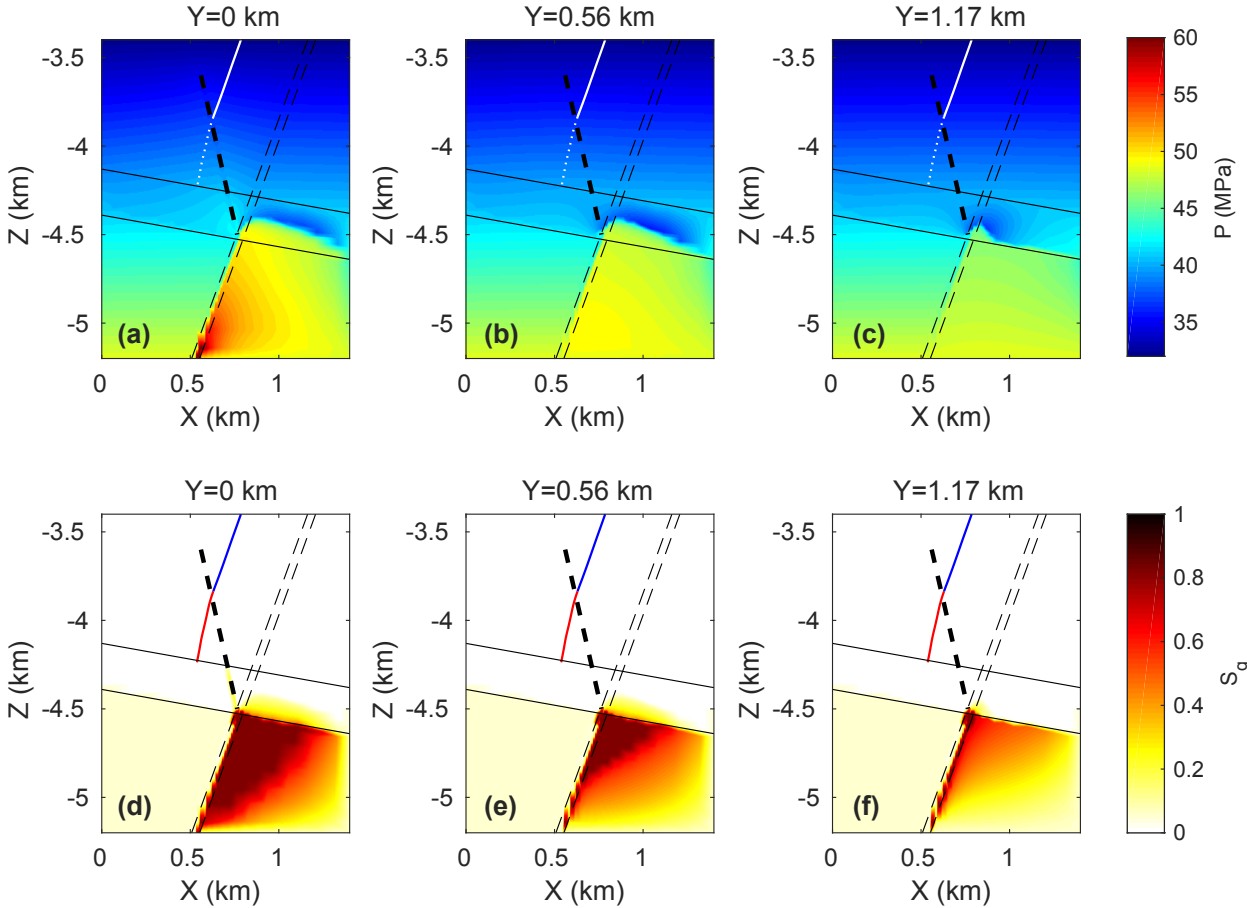

**Figure 6.** Initial conditions of the model: (a–c) Pressure at (a) Y=0 km, (b) Y=0.56 km and (c) Y=1.17 km along profile A–A' in Fig. 2 (normal to the strike of the fault). The well is denoted by the dotted (open section) and solid (cased section) white line. (d–f) Gas saturation at (d) Y=0 km, (e) Y=0.56 km and (f) Y=1.17 km along the profile A–A' in Fig. 2. The well is denoted by the red (open section) and blue (cased section) lines. The fault is illustrated by the thin dashed black lines, the fracture zone by the thick dashed black line, and the caprock by the solid black lines.

### 4.3 Model calibration

We calibrate some of the hydraulic parameters against the bottom-hole and wellhead pressure of the injection test, while the remaining parameters are taken from data or are reasonably assumed from the literature. We use iTOUGH2-PEST (Finsterle and Zhang, 2011) for the calibration according to the approach of a recent study using a coupled hydro-mechanical model

(Rinaldi et al., 2017). The injection test is best suited for model calibration, since the area surrounding the borehole was not affected by previous stimulation. In addition, bottom-hole pressures, probably more representative of the reservoir properties,





**Table 2.** Parameters used for the seed model.

| | |
|---|---|
| Shear modulus $G$ | 4 GPa (used for static stress transfer calculation) |
| Poisson's ratio $\nu$ | 0.25 (used for static stress transfer calculation) |
| Rock density $\rho$ | 2650 kg m$^{-3}$ (Alber and Backers, 2015, and references therein) |
| Cohesion $c$ | 1 MPa |
| Static friction coefficient $\mu_s$ | $0.6 \pm 0.05$ |
| Criticality threshold $\mu_c$ | 0.01 |
| Stress drop coefficient $\Delta\tau_{coeff}$ | 0.05 |
| $b$ value | 1.2 (recalculated from Diehl et al. (2017) for $M_w$) |
| Magnitude of completeness $M_c$ | 0.8 (recalculated from Diehl et al. (2017) for $M_w$) |
| Maximum horizontal stress $S_H$ (Z=-3.8 km) | $134.9 \pm 37.8$ MPa; trend $160 \pm 12°$ (Moeck et al., 2015) |
| Vertical stress $S_v$ (Z=-3.8 km) | $95.7 \pm 2.9$ MPa (Moeck et al., 2015) |
| Minimum horizontal stress $S_h$ (Z=-3.8 km) | $58.4 \pm 7.6$ MPa; trend $70 \pm 12°$ (Moeck et al., 2015) |
| Fault strike $\gamma$ | $210°$ (Diehl et al., 2017) |
| Fault dip $\theta$ | $70°$ (Diehl et al., 2017) |

are only available for the injection test. The pressure data from the acid stimulations are less useful for calibration, since the acidification may have changed the permeability of the carbonate rocks by dissolution, which is not considered in this study. Similarly, pressure data from the gas kick and well control injection are strongly affected by multi-phase fluid interactions, which

prevents a straightforward calibration.

The misfit between the observed and simulated bottom-hole and wellhead pressure data is minimized using a Levenberg-Marquardt algorithm. We put four times more weight on the bottom-hole data, since it is directly coupled to the hydraulic properties of the reservoir. We account for the wellhead pressure to obtain realistic well properties, because the injection occurs in the upper part of the cased borehole. The parameters we invert for are (i) the initial fracture zone permeability, (ii) the

parameter $C_1$ in the permeability-pressure relationship for the fracture zone (Eq. (2)), (iii) the fracture zone compressibility, and (iv) the permeability of the open section of the well.

In order to get a reasonably good fit for the pressure curve, we need to capture the typical fracture opening behavior at $\Delta P \simeq 6$ MPa (sharp kink shortly after the start of injection) as well as the post shut-in behavior that is observed in the data (Fig. 7a). These two observations can be reproduced with an initial fracture zone permeability of $1.3 \cdot 10^{-14}$ m$^2$ that increases

to a maximum permeability of $5.4 \cdot 10^{-13}$ m$^2$ at $\Delta P = 9.4$ MPa ($C_1 = 57.3$, $\alpha = 10^{-8}$) after two hours of injection. Using the cubic law (Witherspoon et al., 1980), these permeabilities correspond to 10 fractures, each with a hydraulic aperture of about 70 $\mu$m (initial) and 235 $\mu$m (fully pressurized). Both the absolute fracture permeability and its changes are within expected ranges derived from in situ data in fractured rock (e.g., Rutqvist, 2015). For the remaining two inversion parameters, we obtain a fracture zone compressibility of $3.7 \cdot 10^{-8}$ Pa$^{-1}$ and a permeability of the open well section of $1.5 \cdot 10^{-8}$ m$^2$. Note that we





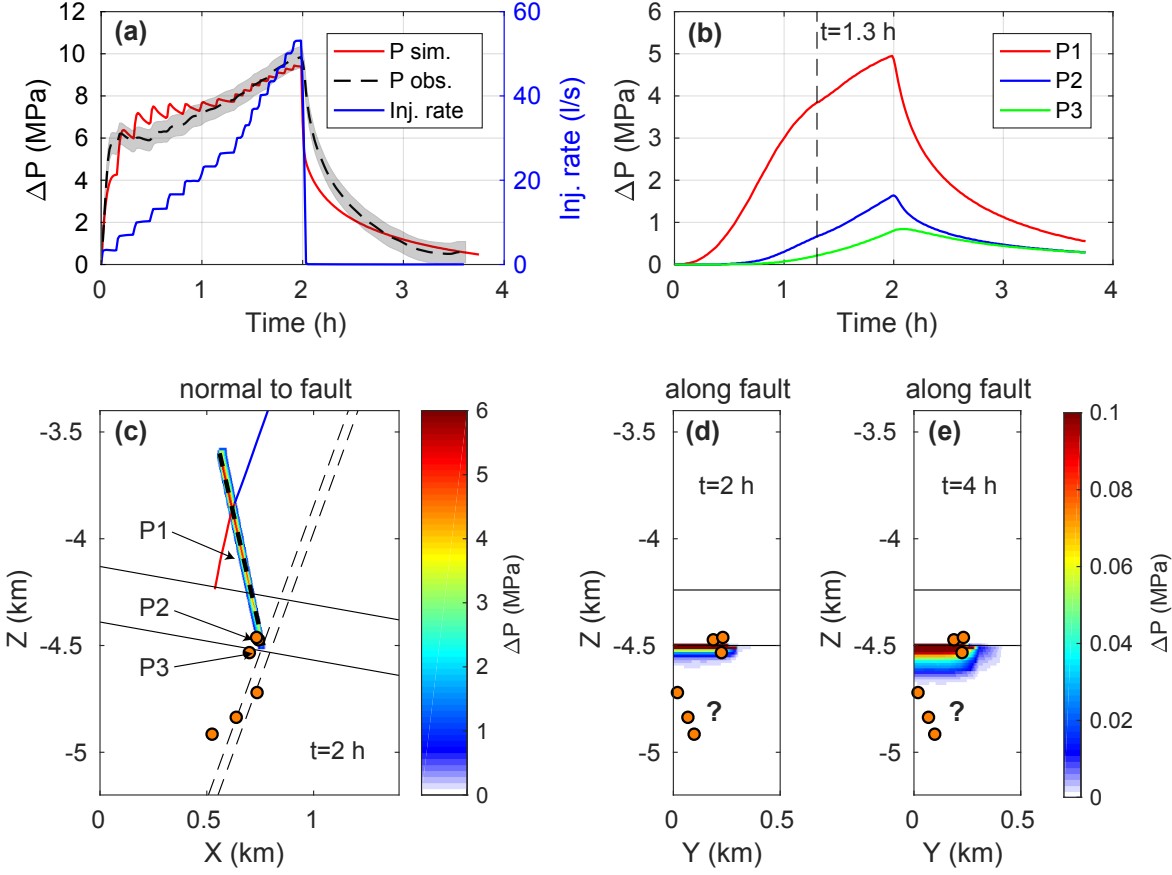

**Figure 7.** Simulated pressure change during the injection test. (a) Comparison of the measured and simulated pressure at the open section of the well after the model calibration with iTOUGH2-PEST. The shaded area denotes an error bound of 0.5 MPa of the measured pressure. (b) Pressure change over time at different points marked in (c): P1 in the fracture zone at a depth of ca. 4.2 km, P2 at the fracture zone/fault intersection in the fracture zone, and P3 at the fracture zone/fault intersection in the fault. The dashed black line marks the onset of seismicity at about 1.3 hours. (c) Pressure change after 2 hours at Y=0 km (along profile A–A' in Fig. 2). The well is denoted by the red (open section) and blue (cased section) lines. The red dots are the projections of the relocated events recorded during the injection test. (d–e) Pressure change after (d) 2 and (e) ca. 4 hours along the fault (left damage zone, along profile B'–B in Fig. 2). The red dots are the projections of the relocated events recorded during the injection test. The deeper three events cannot be explained by a single fracture zone.

can get a similar fit for a different model geometry by adjusting some of the calibrated parameters. We give an example of such a model with two fracture zones below.



## 5 Numerical results and discussion

### 5.1 Injection test

We use the simulation run of the model calibration to quantify the timing and the magnitude of the pressure changes within
the fracture zone and on the fault during the injection test. Figure 7b shows the pressure change over time at different points
in the fracture zone (P1 and P2) and in the fault (P3), whose locations are specified in Fig. 7c. For all three points, a delay and
an attenuation of the pressure response with respect to the well (Fig. 7a) can be observed. While the pressure at the center of
the fracture zone (P1) increases by a maximum of about 5 MPa, the change is only about 1.6 MPa further down in the fracture
zone (P2) and 0.8 MPa on the fault (P3). Despite the delayed response on the fault, an increase of about 0.2 MPa is observed
after 1.3 hours when the seismicity in St. Gallen was initiated (Diehl et al., 2017). Figure 7c illustrates the pressure change and
the relocated seismic events of the injection test on a cross-section normal to the fault (Y=0 km) after 2 hours of simulation
time. The uppermost three events are close to the fracture zone/fault intersection and can be explained by the direct effect of
the pressure caused by the injection, whereas the deeper events are outside the pressurized region. This can also be seen in Fig.
7d and Fig. 7e, where the pressure change and the induced events are shown along the fault after 2 and ca. 4 hours, respectively.
The three deeper events remain outside the pressurized region with $\Delta P \geq 0.1$ MPa until the end of the injection test.

In order to explain the deeper three events, we test a model with a second fracture zone connecting the bottom part of the
well with the fault at a depth of 4.8 km (Fig. 8). Although the quality of the fit is slightly worse compared to the single fracture
zone model, we still reproduce the general trend of the observed pressure (Fig. 8a). Moreover, the pressure increase on the
fault is now significant not only at a depth of 4.5 km, but also at the second fracture zone/fault intersection at a depth of 4.8 km
(Fig. 8b and Fig. 8c). The pressure change on the deeper intersection is smaller compared to the shallower intersection, but the
increase at the onset of seismicity (approx. 0.1 MPa) is still sufficient to induce seismicity. The pressure change along the fault
(Fig. 8d and Fig. 8e) shows that the fault projections of the deeper events lie well within the pressurized regions. Note that the
magnitude of the pressure increase is larger compared to the single fracture zone model (Fig. 7d and Fig. 7e) because the fault
contains less gas and is thus pressurized more due to the absence of a highly compressible fluid phase. As mentioned in Sect.
3, the deeper three events could be a location artifact (Diehl et al., 2017), but if it was not, the model shows that the deeper
three events can be explained by a second fracture zone.

Recently, Zbinden et al. (2019) used a hydro-mechanically coupled model to compare different scenarios with and without
hydraulic connection. In the case of no connection, stress changes on the fault were purely governed by poroelasticity. Although
the results showed that the induced events of the injection test all lay in regions of positive Coulomb stress change (i.e., in
regions promoting failure), the magnitude of the stress changes was in the order of $10^{-3}$ MPa or lower, which is about three
orders of magnitude smaller than in the case of a hydraulic connection (Fig. 7 and Fig. 8). In order to explain the induced
seismicity, Zbinden et al. (2019) concluded that the reactivated fault in St. Gallen was most probably hydraulically connected
to the well. This interpretation was in contrast to another study in which the St. Gallen deep geothermal project was classified
as a site where poroelastic effects, rather than the direct influence of the injection and the associated increase in pore pressure,
predominate at a greater distance from the well (Goebel and Brodsky, 2018). However, in view of our conceptual model with its





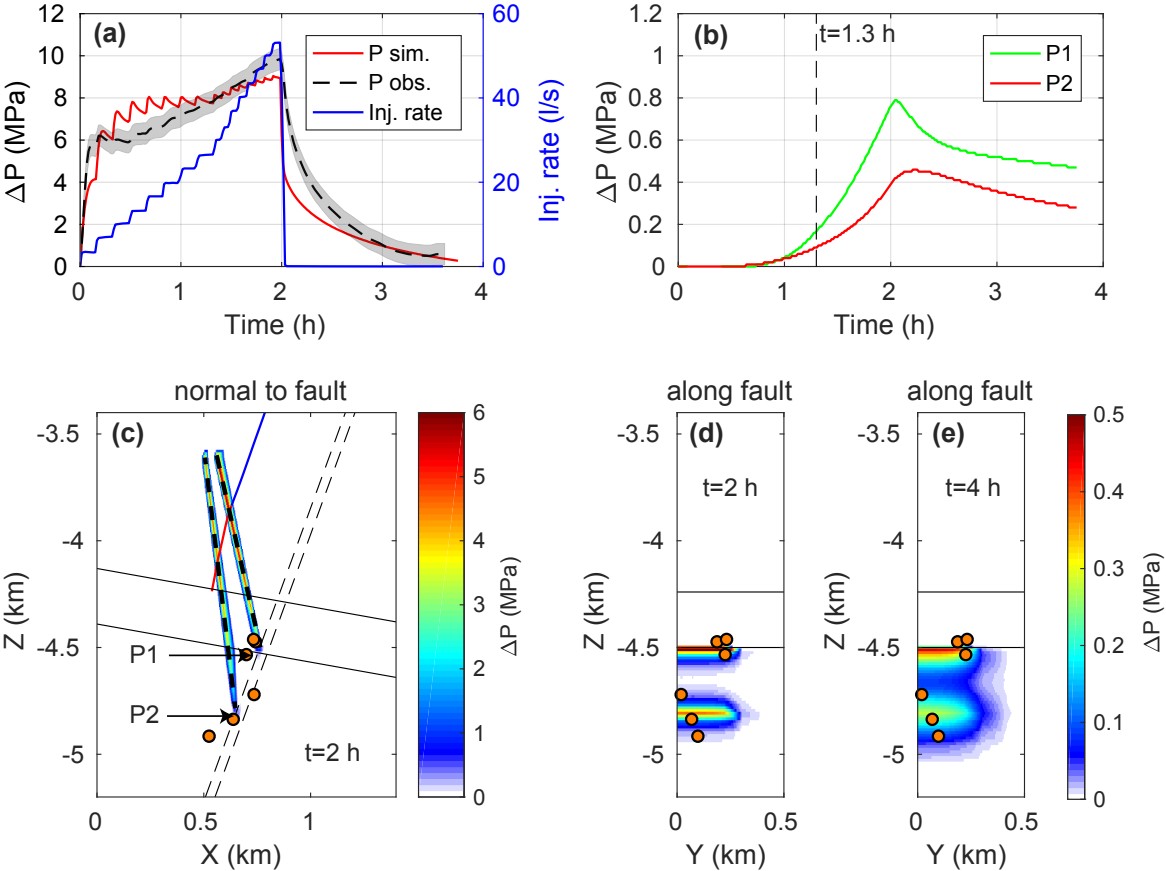

**Figure 8.** Simulated pressure change during the injection test for a model with two connecting fracture zones. (a) Comparison of the measured and simulated pressure at the open section of the well after the model calibration with iTOUGH2-PEST. The shaded area denotes an error bound of 0.5 MPa of the measured pressure. (b) Pressure change over time at different points marked in (c): P1 at the shallower fracture zone/fault intersection and P2 at the deeper fracture zone/fault intersection (both within the fault). The dashed black line marks the onset of seismicity at about 1.3 hours. (c) Pressure change after 2 hours at Y=0 km (along profile A–A' in Fig. 2). The well is denoted by the red (open section) and blue (cased section) lines. The red dots are the projections of the relocated events recorded during the injection test. (d–e) Pressure change after (d) two and (e) four hours along the fault (left damage zone, along profile B'–B in Fig. 2). The red dots are the projections of the relocated events recorded during the injection test.

supporting observations *for* a hydraulic connection and the results from previous simulations (Zbinden et al., 2019), a scenario without hydraulic connection is much less plausible than the fracture zone scenarios examined here.

The results show that the small injection volume of 175 m$^3$ during the injection test is sufficient to cause a significant increase in pore pressure on the distant fault, and can thus promote fault reactivation. Although not modeled here, the same mechanism may have occurred for the acid stimulations, where more fluid was injected (roughly 290 m$^3$; Alber and Backers, 2015, and






references therein) and hence more seismicity was induced. The strong pressure increase during the injection test raises the
question why the connecting fracture zone was not reactivated itself, since in our model, it underwent pressure changes of
several MPa. On the one hand, a higher shear strength (i.e., higher friction or cohesion) of the fracture zone may be sufficient
to prevent reactivation. Moreover, stress modeling has shown that the differential stress acting on the weaker Keuper, Lias and

Dogger formation, where the fracture zone mainly cuts through, may be lower than in other formations (Hergert et al., 2015).
Hence, the fracture zone may be less critically stressed than the reactivated fault. On the other hand, we cannot exclude that
the fracture zone was indeed reactivated, but underwent only aseismic deformation that did not induce any seismic events.

In addition to the potential to reactivate a fault several hundreds of meters away from the well, such a hydraulic connection
could also significantly affect the flow conditions during the operation of a geothermal power plant. If fluid were extracted

from the well, the produced fluid would mainly be governed by inflow from the most permeable structure, i.e., from the highly
permeable fracture zone in our model. The porosity, which determines the amount of fluid contained in the rock, can be very
low in fractured rock. Thus, if production targets such a fracture zone, high flow rates are unlikely to last long, which is
consistent with the low flow rates measured during the production test in St. Gallen (Wolfgramm et al., 2015). In the case of
a second well drilled for fluid production and using the first well for injection (i.e., a geothermal doublet), the injected fluid

would flow rapidly out of the well into the hydraulic connection. Hence, the open section of the production well would need
to intersect the fracture zone at another location to efficiently produce the injected fluid. In our model, a potential location of
the production well could be close to the fracture zone/fault intersection. However, since the efficiency of a geothermal plant
strongly depends on the temperature of the produced fluid (e.g., Schechinger and Kissling, 2015), which is a function of the
residence time of the fluid in the reservoir, a single, highly permeable flow path between the production well and the injection

well would be far from appropriate as the fluid could not heat up sufficiently and the efficiency of the geothermal plant would
be low.

## 5.2   Gas kick

For the gas kick simulation, we follow the hypothesis that the seismicity caused by the initial stimulations breached a seal to
an overpressurized gas reservoir. In order to initiate the kick, we assume a sudden large change in permeability (from $10^{-22}$

$m^2$ to $10^{-15}$ $m^2$) in the fault core right below the caprock (at a depth of 4.5 km). This enables the gas to pressurize the fracture
zone and the bottom-hole (i.e., the fracture zone/well intersection), where the pressure starts to increase after about two hours
(modeled curve in Fig. 9). We do not intend to completely fit the pressure curve measured in St. Gallen, because our high
permeability approach used for the well does not cover the physics of gas lifting, dynamic pressures and pipe friction, which
can play an important role during a kick (e.g., Pan et al., 2018). Instead, we focus on approximately matching the overpressure

at the bottom-hole and the timing of the gas kick. Since the pressure monitoring tool at the well bottom could not be retrieved
after the gas kick and well control measures (T. Bloch, pers. comm., 7 September 2019), we have to reconstruct the pressure
change over time from the available wellhead pressure $P_{wh}$ and injection data. The gas kick occurred shortly after the release
by the operating crew of some gas that had accumulated in the annulus ($\Delta P \simeq 0.6$ MPa). After closing the well, the pressure
rapidly increased to 8 MPa, then slowly rose to about 9 MPa (Fig. 9). At this point, the well was partially gas-filled and the





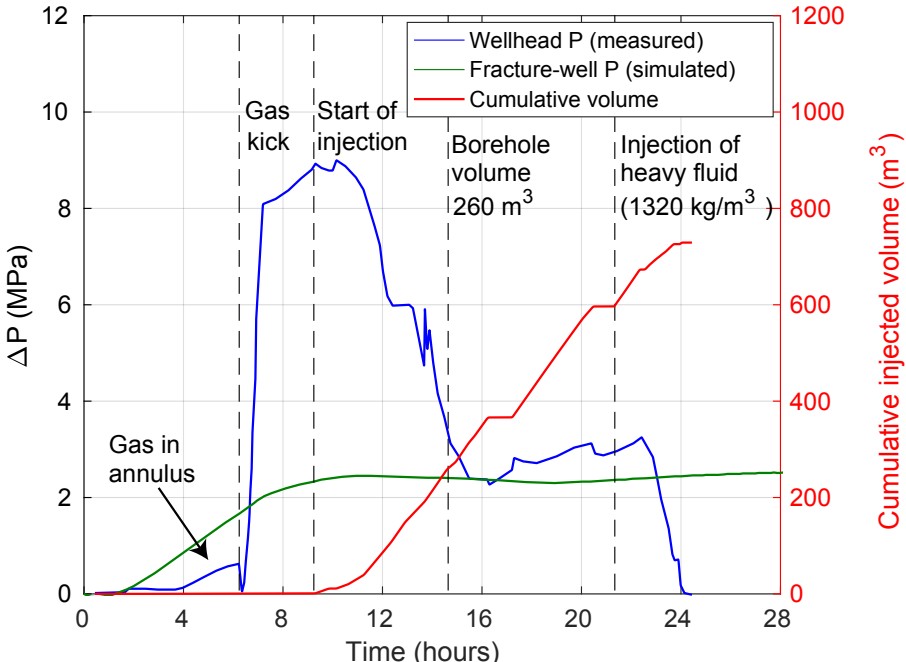

**Figure 9.** Measured wellhead pressure, simulated bottom-hole pressure (fracture zone/well intersection inside fracture zone) and cumulative injected volume with time during the gas kick and well control injection. The vertical dashed black lines mark important events during the well operation (see text for more detailed explanations). The simulated overpressure at the bottom-hole does not exceed the wellhead pressure prior to the injection of heavy fluids. Zero-time corresponds to the breach of the seal to initiate the gas kick in the simulation. Note that the injection is not modeled here.

weight of the fluid column was therefore below the hydrostatic equilibrium. During the injection of fresh water to combat the gas kick, the overpressure dropped to a plateau of 3 MPa shortly after the amount of injected water corresponded to the volume of the borehole ($V_{bh} = 260$ m³; Fig. 9). Subsequently, since continuing water injection would not reduce the pressure further, the borehole was probably almost completely water-filled again (i.e., in hydrostatic equilibrium). Therefore, the overpressure at the bottom-hole caused by the gas may have approximately corresponded to the wellhead pressure of the plateau, i.e. about

3 MPa. Since we cannot rule out that the well was still partially filled with gas, this value is assumed to be a maximum overpressure that could have been caused by the gas. For the simulation, the bottom-hole pressure reaches a steady plateau of ca. 2.4 MPa after about 10 hours, which is similar to the value of the reconstructed bottom-hole overpressure (note that injection is not modeled here). The well could finally be killed ($P_{wh} = 0$) by the injection of dense ($\rho_m = 1320$ kg m$^{-3}$) K$_2$CO$_3$-based drilling fluid (Alber and Backers, 2015; Naef, 2015, and references therein). Assuming a fully fluid-filled borehole, a drilling

mud injection volume of $V_m = 100$ m³ (Alber and Backers, 2015, and references therein) and a water density of $\rho_w = 1000$





kg m$^{-3}$, the average fluid density can be calculated as

$$\rho_{avg} = \frac{[V_m \cdot \rho_m + (V_{bh} - V_m) \cdot \rho_w]}{V_{bh}} = 1123 \text{ kg m}^{-3} \tag{7}$$

which results in a pressure of about 42 MPa at a depth of 3.8 km. This corresponds to an overpressure of approximately 5 MPa with respect to the initial undisturbed reservoir pressure, which is indeed sufficient to prevent gas with an overpressure of 3 MPa to enter the well.

The change of overpressure over time at the wellbottom and the timing of the gas kick is dependent on different parameters such as the overpressure of the gas reservoir, the capillary pressure in the fault and in the fracture zone, and the permeability of the breached fault core. Here, we consider the effect of different permeabilities for the breached fault core, as this does not require reinitialization of the gas plume (Fig. 10). For the base case shown in Fig. 9, we set the permeability of the breached seal to $10^{-15}$ m$^2$. Reducing the permeability by one order of magnitude leads to a significantly slower and weaker pressure response at the fracture zone/well intersection, whereas a tenfold increase yields a similar overpressure as for the base case (Fig. 10a). In the latter case, the peak pressure is reached about 0.2 days before, followed by a pressure drop that is not observed for the base case. The permeability also has an effect on the timing of the gas kick, as illustrated in Fig. 10b. In the case of $\kappa = 10^{-14}$ m$^2$, the gas reaches the fracture zone/well intersection after about 0.2 days, whereas it takes 0.26 days and 0.5 days for the medium ($10^{-15}$ m$^2$) and low ($10^{-16}$ m$^2$) permeability cases, respectively. A second increase in gas saturation can be observed for the medium and high permeability cases after about 0.8 days, which occurs when the gas starts to accumulate in the upper part of the fracture zone between a depth of 3.6 and 3.8 km. The change in pressure at the fracture zone/fault intersection that is located in the vicinity of the breached seal follows a similar trend when compared to the pressure simulated at the fracture zone/well intersection (Fig. 10c). Figure 10d shows the gas saturation of the base case at Y=0 km after 0.4 days, i.e., about 3 hours after the gas has reached the well. At this time, the entire fracture zone has a gas saturation of about 30 %.

### 5.3 Well control injection and associated seismicity

Starting from the same conditions as for the gas kick simulation (base case), we start to inject water after about 0.4 days, i.e. about 3 hours after the gas has reached the well – consistent with the observations illustrated in Fig. 9. We then follow the recorded injection protocol, but ignore the injection of heavy mud at the end of the well control exercise. Using a total of 40'000 seeds, we simulate 1000 realizations of the entire sequence for a simulation time of 10 days. During the sequence, we assume further permeability changes in the fault core associated with the $M_L$ 2.1 and $M_L$ 3.5 events, the two largest earthquakes in the sequence. For the sake of simplicity, we set a permeability of $10^{-15}$ m$^2$ after 0.9 days ($M_L$ 2.1) and 1.0 days ($M_L$ 3.5) on rectangular areas around the corresponding hypocenters that may have slipped (Fig. 11). Additionally, we account for the fact that the fracture zone/fault intersection has encountered some stress drop due to the seismicity induced by the previous activities (see Fig. 3) using a slightly higher criticality threshold in this region ($\mu_c = 0.015$).

Figure 11 shows the pressure change on the left damage zone of the fault together with the simulated seismicity of a single model realization at different times. No seismicity is observed before the gas kick, whereas a few events are induced at a depth of about 4.6 km after the $M_L$ 2.1 and immediately before the $M_L$ 3.5 event (Fig. 11a, Fig. 11b, Fig. 11f and Fig. 11g). Shortly



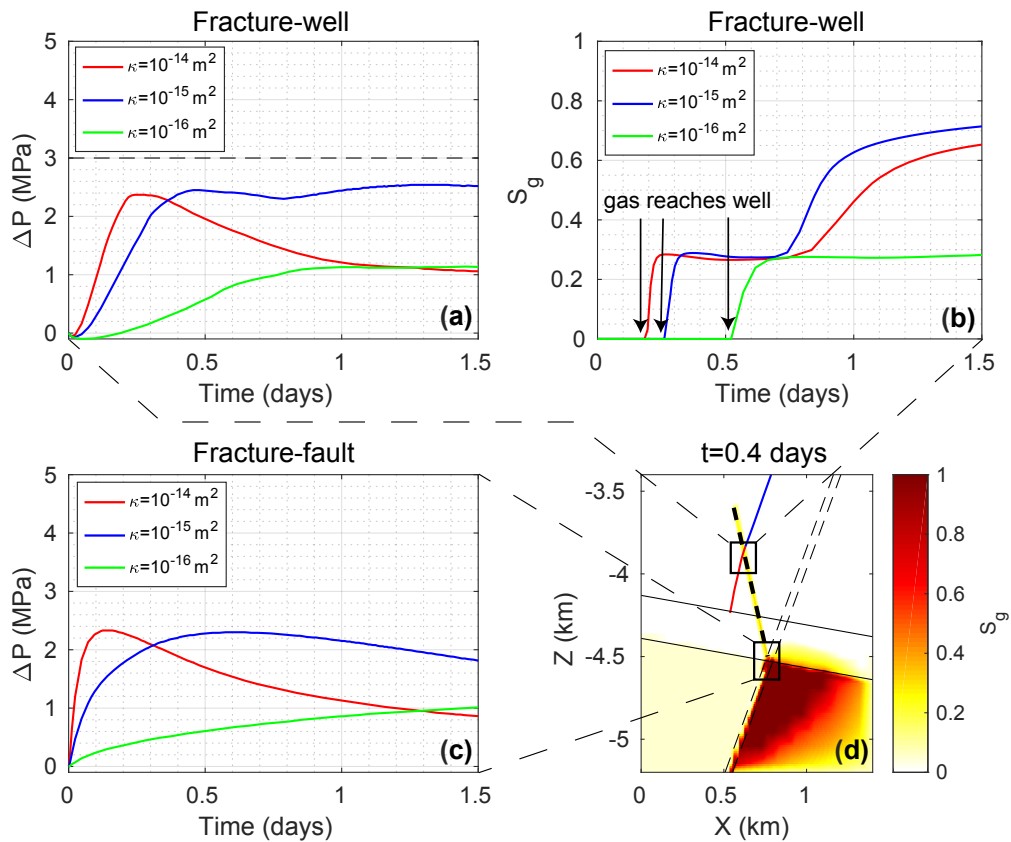

**Figure 10.** (a) Pressure change over time for different permeability values of the broken fault seal at the fracture zone/well intersection (Z≃-3.9 km). The dashed black line at 3 MPa marks the approximate pressure of the plateau in Fig. 9, which is considered as a maximum overpressure that could have been caused by the gas (see text). Zero-time corresponds to the breach of the seal to initiate the gas kick. (b) Gas saturation over time for different permeability values of the broken fault seal at the fracture zone/well intersection (Z≃-3.9 km). (c) Pressure change over time for different permeability values of the broken fault seal at the fracture zone/fault intersection (Z≃-4.5 km, i.e. close to the breached seal). (d) Gas saturation after ca. 0.4 days at Y=0 km (normal to the fault, along profile A–A' in Figure 2).

after the main shock (after 1 day), seismicity has propagated further along the horizontal direction and to greater depth, mainly

in regions where the permeability of the fault core has changed (Fig. 11c and Fig. 11h). During the following days, significantly more seismicity is induced, extending on a patch between a depth of 4.5 and 4.8 km and up to 0.5 km in the Y-axis (Fig. 11d, Fig. 11e, Fig. 11i and Fig. 11j). Regarding the spatial distribution of the seismicity, our model approximately reproduces the extension of the observed seismicity cloud (Fig. 3e and Fig. 3f). The pressure is strongly affected by the permeability changes adopted in the model. The pressure in the left damage zone increases after the $M_L$ 2.1 event in the upper part of the fault (at

a depth of approx. 4.6 km) due to gas inflow, whereas the $M_L$ 3.5 event causes the pressure to decrease in the lower part of the rupture area because water flows across the fault (Fig. 11g and Fig. 11h). On the other hand, the pressure in the fault core

**Figure 11.** (a–e) Pressure change and simulated seismicity (single realization) during the gas kick and well control measures at Y=0 km after (a) ca. 0.4 days (start of injection), (b) 0.99 days (shortly before the $M_L$ 3.5 event), (c) one day (immediately after the $M_L$ 3.5 event), (d) two days and (e) five days of simulation time along profile A–A' in Fig. 2 (normal to the fault). The open section of the well is denoted by the red line. The fault is illustrated by the thin dashed black lines, the fracture zone by the thick dashed black line and the caprock by the solid black lines. (f–j) Pressure change and simulated seismicity (single realization) on the fault (left damage zone) after (f) approx. 0.4 days (start of injection), (g) 0.99 days (shortly before the $M_L$ 3.5 event), (h) one day (immediately after the $M_L$ 3.5 event), (i) two days and (j) five days of simulation time along profile B'–B in Fig. 2 (along fault). The solid black lines indicate the caprock. The gray rectangles denote the area of permeability change to initiate the gas kick (f), due to the $M_L$ 2.1 event (g), and because of the $M_L$ 3.5 event (h).

increases after the main shock, which leads to the reactivation of seeds at greater depth (Fig. 11c). The negative change in pressure is compensated by water and gas inflow, and after 5 days the pressure change becomes positive along the entire fault (Fig. 11j).



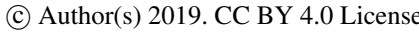

**Figure 12.** Statistical analysis of the main sequence simulation. (a) Number of events for every six hours compared to the entire relocated catalog and events above $M_c$. (b) Cumulative number of events with time compared to the entire relocated catalog and events above $M_c$. (c) Number of events for every six hours induced in total, by the gas and by static stress transfer. (d) Comparison of simulated cumulative number of events with and without stress transfer. The shaded area in (a), (b) and (d) and the error bars in (c) indicate one standard deviation over a total of 1000 realizations. Zero-time corresponds to the breach of the seal to initiate the gas kick.

With the given number of seeds, our model fits the temporal evolution of the seismicity (Fig. 12). Since the relocated catalog is incomplete for events with magnitudes $M_c < 0.8$, we only model events with $M_w \geq M_c$ in order to properly compare the data. We obtain a good match both for the number of events per time interval (Fig. 12a) and for the cumulative number of events (Fig. 12b). For the latter, the observations lie well within one standard deviation of the 1000 realizations of the seed model. Due to the permeability changes in the fault core associated with the two largest seismic events, the simulation nicely captures the





strong increase of seismicity after the main shock at about 1 day. In order to quantify the influence of the gas on the seismicity, we introduce a criterion to assess whether the seeds were triggered by gas or water. We determine a seed to be gas-triggered if the change in gas saturation $\Delta S_g$ with respect to the initial state is positive (i.e., $\Delta S_g > 0$), which is equivalent to $\Delta P > \Delta P_w$ and $\Delta P < \Delta P_g$. A seed is also classified as gas-triggered if the region around it is fully gas saturated and thus $\Delta S_g \geq 0$. If seeds rupture due to static stress transfer, they are classified as neither gas- nor water-triggered. Figure 12c illustrates the gas-

triggered events and the seeds reactivated by static stress transfer in comparison to the total number of events. On average, 39 % of the seeds are triggered by the gas, while another 16 % are triggered by the static stress transfer. Hence, a significant amount of seeds are triggered by an increase in gas pressure. The influence of static stress interactions is more complex to assess, as triggered seeds may cause both negative (stress shadow) and positive Coulomb stress changes on adjacent seeds, bringing them closer or farther away from failure (Catalli et al., 2016; King et al., 1994). Hence, some seeds can be triggered by a change in

fluid pressure (i.e., counted as gas or water-triggered) as a consequence of positive Coulomb stress change, while other seeds can be prevented from failure due to a negative change in the Coulomb stress, although they would have been triggered by fluid flow if stress transfer was ignored. An alternative approach to assess the influence of earthquake interaction is to run a model completely ignoring static stress transfer (Fig. 12d). The result shows that stress transfer starts to play an important role after about 1 day of simulation time (i.e., after the main shock). At the end of the simulation, disregarding stress transfer reduces the

total amount of triggered seeds by about 14 % (note the difference of 2 % to the approach above). Compared to the modeling results obtained for the EGS project in Basel, the influence of stress transfer is less but still of the same order (24 %; Catalli et al., 2016).

     In our simulation, most of the seismicity is induced after the fault core breached during the two main events. In order to determine the effect of the gas and the well control injection on the induced seismicity without the *a priori* assumption of

a breached fault seal, we consider scenarios where permeability is only changed to initiate the gas kick, but not afterwards. In Fig. 13a, we plot the time of the first simulated earthquake (onset) for three different scenarios (1000 realizations each): (i) water injection according to the injection protocol of the well control measures, (ii) no injection, and (iii) opening the wellhead ($P_{wh} = 0.1$ MPa) and no injection. In all three scenarios, the median onset of seismicity occurs a few hours after the time of the actual well control injection. The onset differs only slightly between the three scenarios (0.5 to 0.6 days),

whereas the uncertainty bars given by the first and third quartile show that the seismicity is increasingly delayed without water injection, particularly for the scenario with an open well, where the onset can become larger than 1 day. For the scenarios without injection, in 77 and 96 (open well) realizations no seismicity is induced at all, whereas for the case of injection, in only 21 instances is seismicity absent. In the simulations, although the gas has to cross the fault to cause the gas kick in the well, seismicity does not start immediately after the fault seal breached. This occurs because a certain increase in pressure is required

to reactivate even the most critically stressed seeds (because of the criticality threshold described in Sect. 4.1). The pressure change over time of the analyzed scenarios is shown in Fig. 13b to Fig. 13d. The injection has a large effect at the fracture zone/well intersection, while it is less pronounced further away from the well. At the fracture zone/fault intersection and further down on the fault (at a depth of 4.6 km), the additional pressure increase caused by the water injection is only a few tenths of a MPa. This strong attenuation of the pressure response is mainly caused by the highly compressible gas in the fracture zone and





**Figure 13.** Comparison of scenarios with injection, without injection, and with an open well and no injection (1000 realizations each). (a) Onset of seismicity with errorbars showing the first and third quartile around the median. The dashed line indicates the start of injection (ca. 0.4 days). (b–d) Pressure change at (b) the fracture zone/well intersection, (c) the fracture zone/fault intersection, and (d) within the fault at a depth of 4.6 km. The model sketch on the right shows the locations of the pressure monitoring points.

in the fault that damps the effect of the injection. Opening the well merely has an effect at the fracture zone/well intersection, while no influence can be observed on the fault. Overall, the scenarios suggest that without injection, the seismicity is delayed, but in most cases not absent, which is another indication that the gas significantly affected the induced seismicity. On average, only 4 events are induced for the injection case, whereas less than 3 events occur for the scenarios without injection. These small numbers roughly agree with the relocated catalog, where only 7 events with a magnitude larger than $M_c$ occurred before

the $M_L$ 2.1 event.



### 5.4 Effect of the gas on the induced seismicity

Our simulations suggest that overpressurized gas played an important role during the St. Gallen induced seismicity sequence. Using our definition of gas-triggered events, the simulation that includes permeability changes due to the two main events shows that about 40 % of the seeds were reactivated by the presence of the gas. Note that seeds are not only directly triggered

by an increase in gas pressure, but also indirectly by gas that dissipates the water, which can pressurize more distant regions with unchanged gas saturation. On the other hand, the observed absence of seismicity in the beginning of the gas kick could be seen as an indicator that the gas did not directly affect the induced seismicity. However, our simulation can reproduce the delay of the onset of the seismicity (Fig. 13a) even though the gas has to cross the fault to initiate the gas kick. A delay of about six hours was observed between the onset of the seismicity and the start of the well control injection, as opposed to the injection test

where seismicity begins after one hour. Hence, the reason for the delay could be that the part of the fault that was pressurized first had been reactivated by the previous stimulations and was thus not as critically stressed (as assumed in our model). In the simulation, most of the seismicity occurs after a hydraulic connection between the left and right fault damage zones has been established, implying that these events are induced by multi-phase fluid flow caused by the strong pressure gradient between the two compartments. Indeed, simulations ignoring further permeability changes produced significantly less events

(Fig. 13). Thus, in terms of seismicity, the simulation of the main sequence (Fig. 11 and Fig. 12) including the permeability changes of the two main events corresponds to a worst-case scenario with overpressurized gas. These results are supported by a maximum loss of waveform coherence in the ambient seismic noise field that was observed prior to the main shock, indicating a significant medium change due to fluid flow and associated pore pressure changes possibly caused by overpressurized gas (Obermann et al., 2015). Nevertheless, since most events are induced after the main shock and thus after our assumption of

increased fault core permeability, the effect of the gas on a larger event remains unclear. From a physics-based point of view, our results suggest that the additional pressure increase due to the $M_L$ 2.1 event could have promoted an even larger event, since overpressurized fluid can have a direct influence on the rupture area of induced earthquakes (Galis et al., 2017). Hence, overpressurized gas could explain the large magnitude of the main shock that exceeded the theoretical threshold of the expected maximum magnitude (McGarr, 2014). In this study, because of the random assignment of magnitudes in the model, we cannot

further elaborate on this. A physics-based model accounting for multi-phase fluid conditions and explicitly simulating fault rupture may address this open question (e.g., Zbinden et al., 2017, 2018).

    The extensive permeability change assumed in this model refers to a breach of the fault seal, as suggested by other field and modeling studies (e.g., Lyon et al., 2005; Miller et al., 2004). Additionally, enhanced permeability can be caused by shear dilation of pre-existing fractures (e.g., Lee and Cho, 2002; Rinaldi and Rutqvist, 2019). The sudden change in the hydraulic

properties and associated fluid flow can explain the aftershock sequence of the $M_L$ 3.5 event. However, another important mechanism for aftershocks is static stress transfer, which can lead to stress redistribution around the sliding surface of the main shock, leading to further seismicity (Catalli et al., 2016; King and Devès, 2015). In our simulations, the contribution of static stress transfer is smaller than the influence of the gas (Fig. 12). Király-Proag et al. (2019) have recently analyzed the slip pattern of the main shock by back-projecting relative source time functions of the main events onto the reactivated fault





plane. They found that most aftershocks that occurred within 5 days after the main shock are located at the edge of the $M_L$
3.5 slip area, suggesting that stress concentration due to stress transfer may have played a major role. Here, with the exception
of the assumed permeability changes, we do not explicitly model the main event with the associated stress drop and stress
redistribution. Moreover, we treat fault strength as a static parameter excluding time-dependent failure caused by static stress
transfer. Thus, a detailed geomechanical modeling of the $M_L$ 3.5 event that accounts for time-dependent failure is required to
more accurately quantify the relative contribution of static stress transfer and fluid flow to the aftershock sequence.

The suggested model depends largely upon the initial conditions with a compartmentalized gas reservoir. To further investi-
gate the effect of this assumption, we performed simulations with a model without a fault seal, where the gas reservoir equally
pressurizes the left and right damage zones of the fault (Fig. 14a). In contrast to the case with a sealed fault, in the scenario
without a fault seal the gas kick is initiated after a caprock seal breaks at the fracture zone/fault intersection. The results show
that the gas kick at the bottom-hole is stronger for the unsealed fault (Fig. 14b), because the gas plume is initially located
slightly closer to the fracture zone and the gas does not need to penetrate the broken fault seal (Fig. 14c), which has a tenfold
lower permeability than the rest of the fault. Hence, the gas reaches the well about 0.2 days (approx. 5 hours) earlier compared
to the scenario that includes a fault seal (Fig. 14d). Note that the injection is started 3 hours after the gas kick and thus not
simultaneously for the two scenarios. A comparison of the pressure change at the left damage zone of the fault (at a depth of
4.6 km) shows that the scenarios are inherently different. For the case of a sealing fault core, the pressure increases shortly after
the gas kick is initiated, because the overpressurized gas intrudes a previously undisturbed region (i.e., $P$ close to hydrostatic,
$S_g \simeq 0.1$). For the case without a seal, the pressure is neither affected by the gas kick nor by fluid the fluid injection, because the
fault is fully gas-filled ($S_g = 1$) and already in an overpressurized condition. Hence, the pressure is largely unaffected during
the gas kick, whereas during the injection, the gas is compressed with almost no effect on the pressure. Due to the near zero
pressure changes, we did not perform simulations with the seed model. The comparison shows that the scenario with a sealing
fault and sudden permeability changes can more accurately describe the seismicity observed in St. Gallen. Nevertheless, the
scenario with an unsealed fault also shows that gas does not necessarily enhance the seismicity, as it can damp the effect of the
fluid injection.

Another possible explanation for the seismicity would be purely poroelastic stress changes without any hydraulic connection.
In such a case, it was shown that stress changes on the fault are much smaller than in the case of a hydraulic connection (Zbinden
et al., 2019), implying that the fault needs to be in a highly critical stress state (i.e., only a few $10^{-3}$ MPa or less away from
failure). In addition to the reasons already provided for the injection test, the following observations would be inconsistent with
a poroelastic scenario:

1. Since there is no connection, the gas would probably be stored in the Malm layer or in the Upper Dogger in the vicinity
of the well (e.g., Wolfgramm et al., 2015), which raises the question of why the gas did not enter the well during drilling
       or shortly after the injection test/acid stimulations.





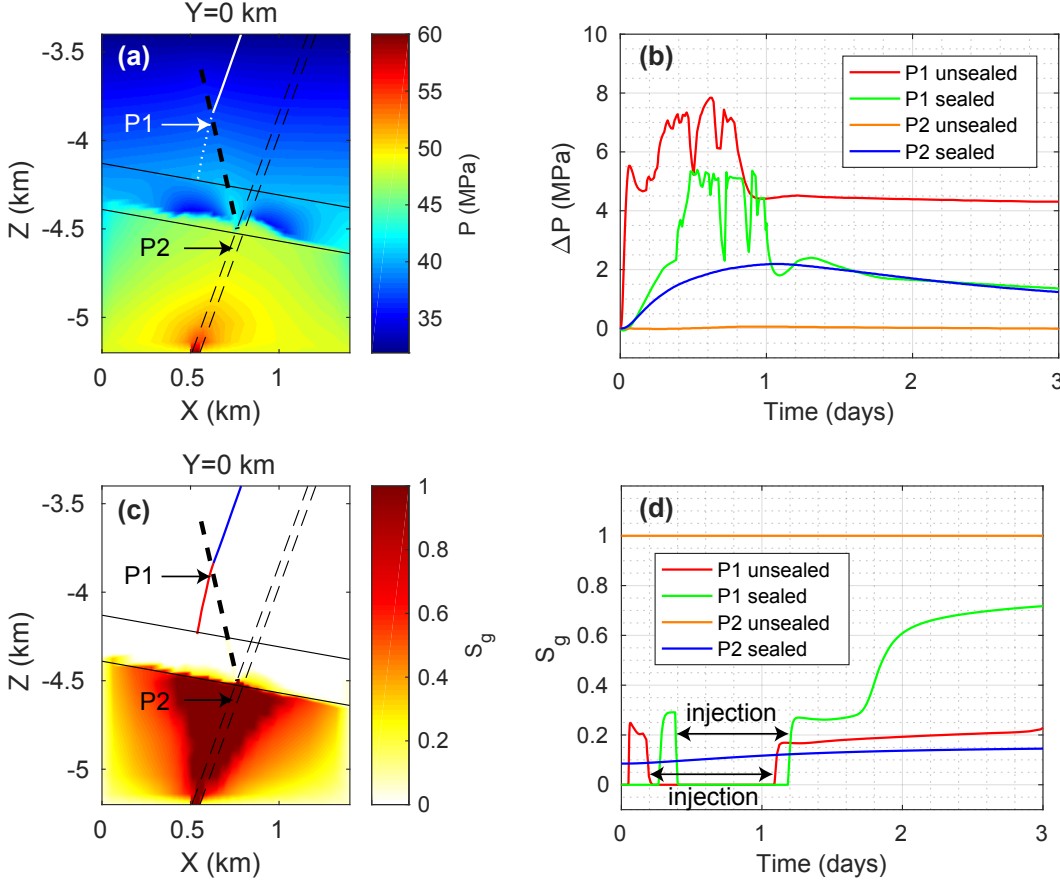

**Figure 14.** Scenario without a fault seal. (a) Initial pressure conditions at Y=0 km (along profile A–A' (normal to the fault) in Fig. 2). The well is denoted by the dotted (open section) and solid (cased section) white line. (b) Pressure change with time at the fracture zone/well intersection and on the fault (Z=-4.6 km) in comparison to the scenario with a sealing fault. (c) Initial gas saturation at Y=0 km (along profile A–A' (normal to the fault) in Fig. 2). The well is denoted by the red (open section) and blue (cased section) line. (d) Gas saturation over time at the fracture zone/well intersection and on the fault (Z=-4.6 km) in comparison to the scenario with a sealing fault. (a, c) The fault is illustrated by the thin dashed black lines, the fracture zone by the thick dashed black line, and the caprock by the solid black lines.

2. The fault is oriented at an angle of about 50° with respect to the maximum principal stress. According to the Mohr-Coulomb theory, an optimally oriented fault would exhibit an angle of 30° (assuming $\mu_s = 0.6$). Hence, the fault in St. Gallen may not be that critically stressed.

3. The seismicity of the post-injection period (September to October 2013) shows diffusion-like propagation characteristics (Diehl et al., 2017) commonly observed for fluid flow.





For these reasons, we did not attempt a model simulation that accounted for poroelastic stress changes. Due to the complex interaction of multi-phase fluid flow and seismicity at St. Gallen, we cannot rule out the possibility that the gas was stored at a different location and was therefore not directly linked to the seismicity. For instance, the gas could have been stored halfway
between the well and the fault, implying that the gas would not directly pressurize the fault during the gas kick. Still, the scenario simulated here is the most likely from our point of view.

## 5.5   Implications for future deep hydrothermal projects

In Switzerland and elsewhere, target reservoirs for future hydrothermal projects may be located at a similar depth and in similar geological conditions as the St. Gallen region. For instance, ongoing projects in the Western Alpine Molasse Basin (WAMB)
at the border between France and Switzerland aim to generate heat and electricity using the high geothermal potential of the area (Chelle-Michou et al., 2017). Due to the similar stratigraphy and the potential presence of permo-carboniferous troughs serving as source rocks for gas, the probability of encountering gas during the drilling and stimulation of deep wells could be relatively high. The simulations on fault seal architecture (Fig. 14) indicate that the effect of gas on the induced seismicity highly depends on the location of the plume and the initial pressure and gas saturation of the fault. Although it is difficult to
accurately evaluate potential locations of gas prior to any stimulation, the presence of a PCT can be a strong indication of gas and should be identified as early as possible in the project, for instance with an active 3-D seismic survey (Heuberger et al., 2016) and gravimetric methods (e.g., Altwegg et al., 2015) as was done in the St. Gallen region. Furthermore, if natural gas is expected to be present at the reservoir depth, it is important to record parameters such as multi-phase flow rates in the well and bottom-hole pressure during the entire project to enable a more accurate hydro-mechanical analysis in both real-time and
retrospectively. In St. Gallen, pressure at the well bottom was recorded with a memory tool, which unfortunately could not be retrieved after the well control measures. With regard to fluid injection, although our simulations strongly suggest that the effect of the well control injection was damped by the compressible gas in the fracture zone and the fault (even in the scenario of a compartmentalized gas reservoir), uncontrolled injection in future projects may greatly increase seismicity and the likelihood of inducing a felt event, and should therefore be avoided where possible. In St. Gallen, stopping the injection of the well control
operation during the gas kick was not an option due to safety reasons at the drilling site.

The simulation of the injection test shows that a hydraulic connection can lead to rapid pressure changes several hundreds of meters away from the well. Since this can lead to problems in the efficient operation of a geothermal power plant (see Sect. 5.1), an option for future hydrothermal projects may for instance be to seal such permeable structures with hydraulic packers. This may provide more uniform flow during injection and production operations and may increase the residence time
of injected fluids, meaning that they can be extracted at a higher temperature from a nearby production well. The installation of packers during the stimulation phase could also help to more locally (i.e, more accurately) characterize the hydraulic properties of the reservoir. Hydraulic testing of small intervals in the open section of the borehole (e.g., individual fractures and faults intersecting the borehole) could significantly improve hydromechanical analyses, and would provide more detailed input data for numerical models. Although expensive, such a multi-stage approach was recently carried out in an EGS project in Finland
(e.g., Kwiatek et al., 2019) and could potentially also be used for hydrothermal systems.



In St. Gallen, the seismicity ceased during a production test conducted in October 2013. While long-term fluid production from porous reservoirs may lead to compaction and induce seismicity on faults intersecting the reservoir (e.g., van Thienen-Visser and Breunese, 2015; Zbinden et al., 2017), in the short-term, production associated with a decrease in pore pressure can increase the effective normal stress and thus stabilize faults close to the reservoir. Moreover, in the case of a fractured low-porosity reservoir like St. Gallen, poroelastic compaction effects may be much less pronounced (e.g., Moeck et al., 2015). Hence, an initial production phase prior to any fluid injection operation may be a strategy to reduce the potential for fault reactivation in such hydrothermal reservoirs.

Finally, since faults can act as conduits for hot fluids from greater depth and can cause positive temperature anomalies (e.g., Chelle-Michou et al., 2017), they are often considered beneficial for geothermal heat production. Such temperature anomalies can be enhanced if a PCT is present, as the rising fluids may form convective cells in the permeable sediments of the permo-carboniferous grabens (Chelle-Michou et al., 2017). Nevertheless, operators must be aware that drilling and injecting into fault zones is always associated with a high probability of inducing felt seismicity, especially in the case of overpressurized gas as shown by the project in St. Gallen.

## 6 Conclusions

We have performed a detailed hydro-mechanical analysis of the multi-phase fluid processes and the induced seismicity at the St. Gallen deep geothermal project with the following results:

1. Based on borehole logs, a seismic survey and the earthquake catalog, we have developed a conceptual model that suggests a highly permeable connection between the injection well and the reactivated fault that is located at several hundreds of meters distance from the well.

2. We implement our concept in a numerical model that is calibrated against the measured well pressure of the injection test. The model shows that the small fluid volumes of the injection test ($175 \text{ m}^3$) are sufficient to yield a fast and significant pressure increase on the fault ($\Delta P \simeq 0.8$ MPa within 1 hour).

3. We simulate the gas kick using the calibrated model and assuming an overpressurized gas reservoir laterally sealed by the fault and released due to the stimulations. The model reproduces the reconstructed overpressure at the bottom-hole during the gas kick ($\Delta P \simeq 3$ MPa).

4. We are able to reproduce the temporal and spatial evolution of the main seismicity sequence following the well control injection. In order to match the aftershock sequence of the $M_L$ 3.5 event, we assume that the fault core is breached during the two largest events of the sequence, which results in strong pressure gradients and associated multi-phase fluid flow.

5. The simulations show that the gas may have played a major role: based on the assumption of a breached fault and the initial conditions in our model, ca. 40 % of the events were directly induced by the gas, while 16 % were triggered by



static stress transfer. Moreover, simulations without an injection delayed the onset of seismicity, but in most cases still induced seismic events. Whether the gas increased the probability of inducing a larger event remains unclear and a more physics-based model regarding fault rupture could provide an answer.

6. In prospect of future deep hydrothermal projects, an initial phase of fluid production could stabilize faults in the vicinity of the reservoir, and thus reduce the potential for fault reactivation. Additionally, since highly permeable fractures intersecting the borehole could become problematic during the long-term operation of a project in terms of heat exchange between the injected fluids and the reservoir rock, hydraulic packers could be used to seal such fractures and thus achieve more uniform flow behavior. Also, prior knowledge of the location of potential source rocks for gas (e.g., PCT) may help
to better assess the hazards not only for gas kicks, but also for induced seismicity. A 3-D seismic survey in combination with other methods (e.g., gravimetric data), as carried out in the St. Gallen area, is therefore strongly recommended prior to any deep hydrothermal project.

*Data availability.* All model output data are available through the ETH repository https://doi.org/10.3929/ethz-b-000369225. The catalog of the induced seismicity in St. Gallen can be found in the supporting information of Diehl et al. (2017). Injection and pressure data of the
injection test are presented in Alber and Backers (2015), and references therein, and can be requested from St.Galler Stadtwerke. Stress data is presented in Moeck et al. (2015) and can be requested from St.Galler Stadtwerke and Professor I. Moeck at the Leibniz Institute for Applied Geophysics (LIAG) in Hannover, Germany.

*Author contributions.* D. Zbinden and A. P. Rinaldi conceptualized the model. D. Zbinden processed the model code and performed the numerical simulations. D. Zbinden prepared the manuscript with contributions from all co-authors.

*Competing interests.* The authors declare that they have no conflict of interest.

*Acknowledgements.* We thank T. Bloch from St.Galler Stadtwerke (sgsw) for his valuable comments that greatly improved the manuscript, and for the permission to use the pressure and injection data as well as supporting documents. We are also grateful to Professor I. Moeck at the Leibniz Institute for Applied Geophysics (LIAG) in Hannover, Germany, for providing estimations on the in situ state of stress. This work was supported by a Swiss National Science Foundation (SNSF)-Ambizione Energy grant (PZENP2_160555).





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
