# Peer review of "Potential influence of overpressurized gas on the induced seismicity in the St. Gallen deep geothermal project (Switzerland)"

_Solid Earth, 2019_

## Referee Comment (RC1) · Anonymous Referee #1 · 19 Jan 2020

Zbinden et al. present a modelling study of the induced seismicity triggered during stimulation of the St Gallen EGS project, Switzerland. The study applies methods and models developed by the authors in previous work. Therefore, the novelty here is in the specific application (St Gallen) and its idiosyncrasies (involvement of gas). The primary finding is to confirm a hypothesised conceptual model using a numerical model that approximates multi-component (water and gas) fluid flow and seismicity triggering (a stochastic "seed" model). The model has difficulty capturing all complexities associated with the stimulation, e.g., borehole processes, breaching of the fault seal, but these are acknowledged and discussed by the authors. Overall, I think is a well-executed study, technically sound and fairly presented. I have listed below a few technical and editorial

[Figure]

comments that the authors may wish to consider, although none are major items.

Abstract: final sentence - "important implications" - could be specific about which implications of the study you think are important. L32: "rock-fluid interaction" suggests geochemistry in many circles, when I think you are referring to fluid destabilisation. L39: "recently, a M 5.5..." awkward phrasing L42: would be appropriate to cite McGarr 2014 here L49: "gas kick" introduced without being defined - an early definition would aid readability L61: very pedantic but "secondly" is not a word IMO. However, if you're going to use the 'ly' then be consistent (e.g., firstly in prev sentence) L271: distribution for coefficient of friction is quoted but not for other parameters in seed model L311: I don't think this is explicitly mentioned - is the gas modelled in TOUGH2 methane or air? L478: With arbitrarily seeded stochastic simulations, your ability to "reproduce the extension (extent) of the observed seismicity" can be challenged as simply a random feature of the realisation. Have you run multiple realisations and confirmed that the observed extent falls within the modelled distribution? L489: pedantic maybe, but "nicely" is perhaps a value judgment best left to the reader

Finally, one thing I missed in the discussion was some comment on model uniqueness. A lot of choices have to be made about parameter values in your model. Even if these are the best, most defensible values, they could still be wrong. Which parameter values do you feel are least well constrained? If the model was rerun (incl. recalibrated) using different plausible values, would you arrive at similar conclusions (either qualitative or quantitative)?
* * *

---

## Referee Comment (RC2) · Anonymous Referee #2 · 19 Mar 2020

In the manuscript entitled "Potential influence of overpressurized gas on the induced seismicity in the St. Gallen deep geothermal project (Switzerland)", the authors conducted a detailed study on induced earthquakes in a geothermal project during which gas kick occurred. After a comprehensive introduction on observation data, the authors set up a hydro-mechanical numerical model to compute the stress perturbations caused by operations in different stages, e.g. injection test, acid stimulations, and gas kick and well control. The modeling results support the hypothesis in that unexpected gas kick induced earthquakes with magnitudes up to ML 3.5. Overall the manuscript is well written and I can easily follow the logic. I think the manuscript can be accepted after minor revisions.

[Figure]

In the Introduction the authors list a number of anthropogenic activities that may induce earthquakes. One type of activity, which has direct connection and may benefit from the results of this study, is large underground gas storage (UGS) where cyclic injection and extraction of natural gas is conducted. UGSes have been built globally, with notable examples with large capacity in China. It has been recently reported that the injection and extraction of natural gas in a large UGS may induce earthquakes (Zhou et al., 2019; Jiang et al., 2020).

Zhou, P., H. Yang, B. Wang, and J. Zhuang (2019), Seismological investigations of induced earthquakes near the Hutubi underground gas storage facility, J. Geophys. Res., doi:10.1029/2019JB017360

Jiang, G., X. Qiao, X. Wang, R. Lu, L. Liu, H. Yang, Y. Su, L. Song, B. Wang, and T.F. Wong (2020), GPS observed horizontal ground extension at the Hutubi (China) underground gas storage facility and its application to geomechanical modeling for induced seismicity, Earth Plane. Sci. Lett., 530, https://doi.org/10.1016/j.epsl.2019.115943

Indeed the Hutubi UGS was bounded by different faults, which now seal the reservoir. The reported findings in this study have implications on potential changes on fault permeability by smaller earthquakes and thus causing gas flow/leakage from the reservoir or repository. This can be added in discussion and help expand the horizon.

Is that necessary to add another fracture zone to explain those deeper earthquakes? If using fully coupled poroelastic model, poroelastic stress perturbation would be sufficient to induce earthquakes that were 300 m away. Even for injection of gas, poroelastic stress changes are sufficiently large to induce earthquakes (e.g. Jiang et al., 2020). The argument in lines 385 to 392 seems to draw a conclusion based on the horizontal fracture zone (Fig. 8d&e). While the earthquakes are probably too small to derive focal mechanisms, Coulomb failure stress is quite sensitive to receiver fault geometry. So I do not think the justification here is very convincing. Indeed it is quite common to observe induced earthquakes beneath the injection or extraction zone, depending on

fault orientation.

In the model the fault core is set as 5 m wide low permeability zone. According to observations of exhumed faults, most crustal faults have fault cores in cm scale, where earthquake slip is concentrated. Such 5 m scale is limited by the model, or is intended to set in such a scale? What is the effect of such scale, for example, if you decrease it by one order?

---

## Author Comment (AC1) · 6 Apr 2020

Summary and evaluation by reviewer

Zbinden et al. present a modelling study of the induced seismicity triggered during stimulation of the St Gallen EGS project, Switzerland. The study applies methods and models developed by the authors in previous work. Therefore, the novelty here is in the specific application (St Gallen) and its idiosyncrasies (involvement of gas). The primary finding is to confirm a hypothesized conceptual model using a numerical model that approximates multi-component (water and gas) fluid flow and seismicity triggering (a stochastic "seed" model). The model has difficulty capturing all complexities associated

with the stimulation, e.g., borehole processes, breaching of the fault seal, but these are acknowledged and discussed by the authors. Overall, I think is a well-executed study, technically sound and fairly presented. I have listed below a few technical and editorial comments that the authors may wish to consider, although none are major items.

We would like to thank the reviewer for the positive review and the comments, which were all considered in the updated manuscript. Our detailed response to each comment can be found below. We would like to clarify that, although the St. Gallen deep geothermal project has been considered an EGS in some studies (e.g., Breede et al., 2013), we prefer to classify it as a hydrothermal project, as no hydraulic stimulation for targeted shearing of fractures (i.e., hydro-shearing) adjacent to the injection well was performed.

Breede, K., Dzebisashvili, K., Liu, X. et al (2013). A systematic review of enhanced (or engineered) geothermal systems: past, present and future. Geotherm Energy 1, 4. https://doi.org/10.1186/2195-9706-1-4

(1) Reviewer comments (2) Author response* (3) Changes in the manuscript

*Line numbers refer to the initially submitted manuscript

Detailed Comments:

(1) Abstract: final sentence - "important implications" - could be specific about which implications of the study you think are important.

(2) We now explicitly mention that the study could have implications for future deep hydrothermal projects where potentially overpressurized gas may be in-place.

(3) "This study may have implications for future deep hydrothermal projects conducted in similar geological conditions with potentially overpressurized in-place gas."

(1) L32: "rock-fluid interaction" suggests geochemistry in many circles, when I think you are referring to fluid destabilisation.

(2) We understand that the term "rock-fluid interaction" is not completely clear. Here, we meant any kind of interaction between rock and fluids (water and gas) including thermal, hydraulic, chemical and mechanical processes. Therefore, we now use the term "thermo-hydro-mechanical-chemical interactions" to avoid any confusion. Note that we do not further consider chemical and thermal processes in our simulations, since we neither model the acid stimulations nor expect a significant cooling effect over the relatively short-term and small volume injection that occurred in St. Gallen (L 304-307).

(3) "Hence, it is crucial to get a more accurate understanding of the thermo-hydro-mechanical-chemical interactions occurring at reservoir depths."

(1) L39: "recently, a M 5.5..." awkward phrasing

(2) The reviewer is correct, we have changed the sentence accordingly.

(3) "recently, a Mw 5.5 earthquake struck the city of Pohang (South Korea) (Ellsworth et al., 2019; Grigoli et al. 2018), the largest earthquake recorded at an EGS site up to date (Kim et al., 2018)."

(1) L42: would be appropriate to cite McGarr 2014 here

(2) The reviewer is correct, we cited Eaton and Igonin (2018) that summarized recently proposed approaches to estimate the maximum induced magnitude. We now cite McGarr (2014) and another recently proposed model that relates the total injected volume to the maximum arrested rupture (Galis et al., 2017) instead of Eaton and Igonin (2018).

(3) "This earthquake has challenged recently proposed models that relate the maximum expected seismic magnitude to the total injected fluid volume (Galis et al., 2017; McGarr, 2014)."

(1) L49: "gas kick" introduced without being defined - an early definition would aid readability
(2) We now define the term "gas kick" where it is first mentioned in the text.

(3) "..., gas entered the borehole from an unidentified source at a pressure greater than the one exerted by the fluid column in the borehole (a so-called gas kick). The gas kick ..."

(1) L61: very pedantic but "secondly" is not a word IMO. However, if you're going to use the 'ly' then be consistent (e.g., firstly in prev sentence)

(2) We now use "firstly" in the previous sentence to be consistent.

(3) "Firstly, we describe the temporal and spatial evolution of the seismic sequence associated with the injection. Secondly, we present ..."

(1) L271: distribution for coefficient of friction is quoted but not for other parameters in seed model

(2) In the seed model, other parameters, such as the shear modulus of the fault, Poisson's ratio, fault cohesion and stress drop coefficient were constant with no normal distribution around its mean. In addition to the friction coefficient, parameters with a normal distribution were the magnitude of the horizontal and vertical stress and the orientation of the horizontal stress, the latter corresponding to a normally distributed orientation of the fault strike (while the horizontal stress is held at a fixed orientation). Stress values and orientations are quoted in L 323-330, while the seed parameters with their distributions are listed in Table 2.

(3) No changes in the manuscript.

(1) L311: I don't think this is explicitly mentioned - is the gas modelled in TOUGH2 methane or air?

(2) The gas is air, which we now clarify in L 308-309. Methane gas and air (ca. 78 % nitrogen) are both in a supercritical state at reservoir conditions (e.g., Nasrifar and Bolland, 2006), i.e., their dynamic viscosity is similar to a gas (in the order of 1e-5

Pa s) and their densities are between a liquid and a gas (about 150 to 300 kg m-3). Therefore, the use of air instead of methane is an appropriate approximation for the purposes of our study.

Nasrifar, K., and Bolland, O. (2006). Prediction of thermodynamic properties of natural gas mixtures using 10 equations of state including a new cubic two-constant equation of state. Journal of Petroleum Science and Engineering, 51(3-4), 253-266. https://doi.org/10.1016/j.petrol.2006.01.004

(3) "In order to model the multi-phase fluid system, we employ an equation of state with water and air as liquid and gas phase, respectively."

(1) L478: With arbitrarily seeded stochastic simulations, your ability to "reproduce the extension (extent) of the observed seismicity" can be challenged as simply a random feature of the realisation. Have you run multiple realisations and confirmed that the observed extent falls within the modelled distribution?

(2) We thank the reviewer for this comment. In the initially submitted paper, we already accounted for the 1000 realizations, but we did not calculate the average of the spatial distribution of the realizations. We now also quantify the extension of the simulated and observed induced seismicity: the mean extent of the 1000 model realizations is 0.133 km2 with a standard deviation of 0.025 km2, while the area of the observed seismicity cloud is 0.214 km2 (only considering events with a magnitude larger than the magnitude of completeness). Hence, the extent of the observed seismicity is greater than in the simulations, even if one standard deviation is taken into account. We now clearly indicate these results in the manuscript.

(3) "Regarding the spatial distribution of the seismicity, our model approximately reproduces the extension of the observed seismicity cloud (Fig. 3e and Fig. 3f), although the simulated seismicity cloud is somewhat smaller than the observed one: the mean extent of the seismicity of the 1000 model realizations is 0.133 km2 with a standard deviation of 0.025 km2, while the area of the observed seismic events (with magnitudes

greater than Mc, see below) is 0.214 km2."

(1) L489: pedantic maybe, but "nicely" is perhaps a value judgment best left to the reader

(2) We agree with the reviewer, thus we removed the word "nicely".

(3) "..., the simulation captures the strong increase of seismicity after the main shock at about 1 day."

(1) Finally, one thing I missed in the discussion was some comment on model uniqueness. A lot of choices have to be made about parameter values in your model. Even if these are the best, most defensible values, they could still be wrong. Which parameter values do you feel are least well constrained? If the model was rerun (incl. recalibrated) using different plausible values, would you arrive at similar conclusions (either qualitative or quantitative)?

(2) Our model with a hydraulic connection between the injection well and the reactivated fault is based on one of our previous studies (Zbinden et al., 2020). We found that several fracture zone parameters (permeability, porosity, compressibility) affect the pressure response at the well and on the fault. For calibrated models, however, the response in terms of pressure and stress changes was comparable, which leads to similar conclusions. We now write in Section 5.1:

(3) "Zbinden et al. (2020) found that several fracture zone parameters affect the pressure response at the well and on the fault. For calibrated models, however, the response in terms of pressure and stress changes was comparable, thus leading to similar conclusions."

Additional reply to the last reviewer comment:

The properties of the hydraulic connection are also critical for the simulation of the gas kick, the well control and the main sequence of the induced seismicity. There, the most uncertain parameters are the location and overpressure of the gas plume, and the permeability of the breached parts of the fault seal. We argued in L 446-449 that changing the permeability of the breached fault seal is similar to varying the overpressure of the gas reservoir, as both influence the timing and strength of the gas kick (effect of permeability of the breached fault seal on pressure evolution is shown in Fig. 10 in the manuscript). One different location of the gas plume was tested in a scenario without a fault seal (Fig. 14), and further possible scenarios were discussed in L 602-606. We mentioned that if the gas was stored elsewhere, it may not be directly linked to the seismicity. However, given the stratigraphy and the observed delay between the stimulations and the gas kick, we consider it most likely that the gas was stored in the permo-carboniferous trough and in the Muschelkalk layer (L 161-172, Fig. 3 and 4).

―――――――――――――――――――――

---

## Author Comment (AC2) · 6 Apr 2020

Summary and evaluation by the reviewer

In the manuscript entitled "Potential influence of overpressurized gas on the induced seismicity in the St. Gallen deep geothermal project (Switzerland)", the authors conducted a detailed study on induced earthquakes in a geothermal project during which gas kick occurred. After a comprehensive introduction on observation data, the authors set up a hydro-mechanical numerical model to compute the stress perturbations caused by operations in different stages, e.g. injection test, acid stimulations, and gas kick and well control. The modeling results support the hypothesis in that unexpected

gas kick induced earthquakes with magnitudes up to ML 3.5. Overall the manuscript is well written and I can easily follow the logic. I think the manuscript can be accepted after minor revisions.

We are glad that the reviewer found our paper interesting and are grateful for the comments, which helped to improve the manuscript. Please find our detailed reply to each comment below.

(1) Reviewer comments (2) Author response (3) Changes in the manuscript

Detailed Comments:

(1) In the Introduction the authors list a number of anthropogenic activities that may induce earthquakes. One type of activity, which has direct connection and may benefit from the results of this study, is large underground gas storage (UGS) where cyclic injection and extraction of natural gas is conducted. UGSes have been built globally, with notable examples with large capacity in China. It has been recently reported that the injection and extraction of natural gas in a large UGS may induce earthquakes (Zhou et al., 2019; Jiang et al., 2020).

Zhou, P., H. Yang, B. Wang, and J. Zhuang (2019), Seismological investigations of induced earthquakes near the Hutubi underground gas storage facility, J. Geophys. Res., doi:10.1029/2019JB017360

Jiang, G., X. Qiao, X. Wang, R. Lu, L. Liu, H. Yang, Y. Su, L. Song, B. Wang, and T.F. Wong (2020), GPS observed horizontal ground extension at the Hutubi (China) underground gas storage facility and its application to geomechanical modeling for induced seismicity, Earth Plane. Sci. Lett., 530, https://doi.org/10.1016/j.epsl.2019.115943

(2) In the introduction, we now mention UGS as an additional industrial application that can induce earthquakes.

(3) "Anthropogenic earthquakes have been observed related to water impoundment, mining, geothermal power production, hydrocarbon extraction, hydraulic fracturing for

shale gas extraction, $CO_2$ sequestration, wastewater injection, and cyclic injection and extraction operations at underground gas storage (UGS) sites (Ellsworth, 2013; Grigoli et al., 2017; Foulger et al., 2018)."

(1) Indeed the Hutubi UGS was bounded by different faults, which now seal the reservoir. The reported findings in this study have implications on potential changes on fault permeability by smaller earthquakes and thus causing gas flow/leakage from the reservoir or repository. This can be added in discussion and help expand the horizon.

(2) We thank the reviewer for the abovementioned papers, which are now cited in the manuscript. Additionally, we briefly discuss the implications of fault seal permeability changes for UGS sites, in particular for the HUGS, where the seals of the bounding faults could be breached due to the induced seismicity and thus cause gas leakage.

(3) "Such permeability changes in sealing faults due to induced seismicity can have implications for other geo-energy applications, such as $CO_2$ sequestration and UGS. For instance, the large Hutubi underground gas storage (HUGS) facility in northwestern China is bound by multiple faults sealing the reservoir (Jiang et al., 2020). These seals may be damaged by small induced earthquakes reported in the field (Zhou et al., 2019), which could cause gas leakage."

(1) Is that necessary to add another fracture zone to explain those deeper earthquakes? If using fully coupled poroelastic model, poroelastic stress perturbation would be sufficient to induce earthquakes that were 300 m away. Even for injection of gas, poroelastic stress changes are sufficiently large to induce earthquakes (e.g. Jiang et al., 2020). The argument in lines 385 to 392 seems to draw a conclusion based on the horizontal fracture zone (Fig. 8d&e). While the earthquakes are probably too small to derive focal mechanisms, Coulomb failure stress is quite sensitive to receiver fault geometry. So I do not think the justification here is very convincing. Indeed it is quite common to observe induced earthquakes beneath the injection or extraction zone, depending on fault orientation.

(2) Lines 385 to 392 discuss a possible alternative scenario without a fracture zone, which has been investigated in one of our previous studies (Zbinden et al., 2020). Based on a fully physics-based hydro-mechanical model, we concluded that a hydraulic connection between the fault and the injection well is a more plausible scenario for the St. Gallen case, because stress changes purely governed by poroelasticity were too small to induce the seismicity. Note that the magnitude of poroelastic stress changes not only depends on the distance between the fault and the well, but also on the total injected fluid volumes, which were rather low at St. Gallen. We agree with the reviewer that adding a second fracture zone to explain the three deeper events induced during the injection test is unnecessary, because (i) it does not affect any of our conclusions, and (ii) Diehl et al. (2017) argued that the location of the deeper events is most probably an artifact caused by a local vp/vs velocity anomaly and thus these seismic events would be located at shallower depth. This would allow to explain all the relocated events induced by the injection test with only one fracture zone. Since our paper is already relatively long, we decided to completely remove the scenario with a second hydraulic connection (including Fig. 8) from the paper. We then changed the manuscript as follows:

(3) In Section 3: "The second hydraulic connection could then explain the fast seismic response to the stimulations in the lower part of the fault. However, despite these observations, Diehl et al. (2017) proposed that the vertical offset of this cluster is a location artifact, which can be explained by the presence of a local vp/vs velocity anomaly. For this reason, we choose to perform the numerical simulations with only one hydraulic connection."

In Section 5.1: "The three deeper events could be explained by a second fracture zone connecting the well with the reactivated fault at greater depth. However, as mentioned in Sect. 3, the location of the deeper events is most probably an artifact (Diehl et al., 2017), which would allow to explain all the induced events with only one fracture zone."

(1) In the model the fault core is set as 5 m wide low permeability zone. According to

observations of exhumed faults, most crustal faults have fault cores in cm scale, where earthquake slip is concentrated. Such 5 m scale is limited by the model, or is intended to set in such a scale? What is the effect of such scale, for example, if you decrease it by one order?

(2) The thickness of 5 m corresponds to the width of the fault core elements. However, we do not think that fault core thickness is a crucial parameter in our model. Firstly, for the initialization of the gas plume, the fault seal is not only a hydraulic seal (i.e., very low permeability), for which the thickness would have an effect, but also a membrane seal (e.g., Yielding et al., 1997) due to the high capillary entry pressure adopted in the fault core. Hence, the fault seal could maintain a similar overpressure of the gas reservoir even with a smaller thickness. Secondly, we would indeed expect some influence of fault core thickness on the strength of the gas kick, because reducing the thickness would result in a higher pressure gradient between the two reservoir compartments, which would cause more fluid flow across the fault after the seal has been breached. However, according to Darcy's law, this would be equivalent to increasing the permeability of the breached fault seal (which would also result in more flow). This is exactly what we did in Fig. 10, where we show the effect of the permeability of the breached fault seal on the pressure and gas saturation change at the fracture-well and fracture-fault intersections. Decreasing the fault core thickness by one order would thus correspond to the scenario with a breached fault seal permeability of 10-14 m2. We now explain this equivalency in the manuscript in Section 5.2:

Yielding, G., Freeman, B., and Needham, D. T. (1997). Quantitative fault seal prediction. AAPG Bulletin, 81(6), 897-917.

(3) "Note that according to Darcy's law, an increase (decrease) in permeability of the breached fault core would correspond to a decrease (increase) in thickness of the fault core, since in both cases the fluid flow across the fault would be equally affected."

Additional reply to the last reviewer comment:

Although fault core thickness can vary over a wide range, we agree with the reviewer that individual fault cores are usually thinner than 1 m (e.g., Shipton et al., 2006). However, faults may contain multiple narrow cores (e.g., Faulkner, 2010) so that the sealing part of the fault can be thicker. Such approximations are often used in numerical modeling studies, where fault seals with thicknesses up to several meters have been adopted (e.g., Rinaldi et al., 2014; Jiang et al., 2020). Furthermore, we do not explicitly calculate fault slip for individual earthquakes. Therefore, from a mechanical modeling perspective, the thickness of the fault core is not important here. Nevertheless, even if fault slip is taken into account, it was shown that the finite-thickness approach leads to similar results compared to using a zero-thickness interface approach for the fault (Cappa and Rutqvist, 2011).

Cappa, F., and Rutqvist, J. (2011). Modeling of coupled deformation and permeability evolution during fault reactivation induced by deep underground injection of CO2. International Journal of Greenhouse Gas Control, 5(2), 336-346. https://doi.org/10.1016/j.ijggc.2010.08.005

Faulkner, D. R., Jackson, C. A. L., Lunn, R. J., Schlische, R. W., Shipton, Z. K., Wibberley, C. A. J., and Withjack, M. O. (2010). A review of recent developments concerning the structure, mechanics and fluid flow properties of fault zones. Journal of Structural Geology, 32(11), 1557-1575. https://doi.org/10.1016/j.jsg.2010.06.009

Rinaldi, A. P., Rutqvist, J., and Cappa, F. (2014). Geomechanical effects on CO2 leakage through fault zones during large-scale underground injection. International Journal of Greenhouse Gas Control, 20, 117-131. https://doi.org/10.1016/j.ijggc.2013.11.001

Shipton, Z.K., Soden, A.M., Kirkpatrick, J.D., Bright, A.M., and Lunn, R.J (2016). How thick is a fault? Fault displacement-thickness scaling revisited. In Abercrombie, R. (Eds) Earthquakes: Radiated Energy and the Physics of Faulting, pp. 193-198. AGU.

---

## Author Response (AR2)

**REPLY TO EDITOR CORRECTIONS**
D. Zbinden, A. P. Rinaldi, T. Diehl, S. Wiemer

**"Potential influence of overpressurized gas on the induced seismicity in the St. Gallen deep geothermal project (Switzerland)"**

submitted to *Solid Earth*
[MS No.: se-2019-156]

(1) Editor comments
(2) Author response
(3) Changes in the manuscript

*Tarje Nissen-Meyer (Topical Editor)*

Dear authors,

Many thanks for your detailed and attentive replies and updated manuscript. I am now happy to accept your manuscript for publication in SE after inspecting your changes. I suggest technical corrections only very minor suggestions, but leave it to your valued judgment to update these:

We are grateful to Tarje Nissen-Meyer for the positive assessment of the manuscript and the technical corrections. Please find a detailed response to these comments below.

If a reviewer misses the description of parameters (only friction coefficient in seed model), chances are that others may overlook it similarly. I suggest mentioning around the quoted section where you describe the other parameters.

We moved the sentence describing the exact values for the friction coefficient from Sect. 4.1 to 4.2. Moreover, we now clearly write that friction and the state of stress are the only parameters in the seed model that follow a normal distribution, while all other parameters are constant. We now write in Section 4.2:

"We assume the seeds to have a coefficient of friction of $0.6 \pm 0.05$ and a cohesion of 1 MPa. … Note that all parameters in the seed model, except for the coefficient of friction and the state of stress that follow a normal distribution, are assumed to be constant. A list of the seed model parameters is given in Table 2."

I would find it insightful to refer to the Nasrifar paper for those who wish to follow up on the choice of the gas model

We now more accurately motivate the choice of the gas model and refer to Nasrifar and Bolland (2006).

"Methane and nitrogen (air contains approx. 78 % nitrogen by volume) are both in a supercritical state at reservoir conditions (e.g., Nasrifar and Bolland, 2006), i.e., their dynamic viscosity is similar to a gas and their density is between a liquid and a gas. We therefore consider the use of air instead of methane to be an appropriate approximation for the purposes of this study."

Please use "extent" instead of "extension of observed seismicity" as suggested.

Corrected.

"… our model approximately reproduces the extent of the observed seismicity cloud …"

I appreciate the detailed explanation of the 5m fault core thickness, and wouldn't mind seeing a bit more of this in the updated manuscript. I believe readers may benefit from this discussion.

We now discuss in more detail the choice of fault core thickness and its influence on the strength of the simulated gas kick. In Sect. 5.2, we write:

"Similar to the permeability of the breached fault seal, we expect the thickness of the fault core to influence the strength of the gas kick, because reducing the thickness would result in a higher pressure gradient between the two reservoir compartments, which would cause more fluid flow across the fault after the seal has been breached. In our numerical model, the thickness of 5 m corresponds to the width of the fault core elements. Individual fault cores are usually thinner than 1 m (e.g., Shipton et al., 2006), but faults may contain multiple narrow cores (e.g., Faulkner, 2010) so that the sealing part of the fault can be thicker, which can justify our assumption in the model. Moreover, note that according to Darcy's law, an increase (decrease) in permeability of the breached fault core would correspond to a decrease (increase) in thickness of the fault core, since in both cases the fluid flow across the fault would be equally affected. For instance, decreasing the fault core thickness by one order would correspond to the scenario with a breached fault seal permeability of $10^{14}$ m2. Hence, the sensitivity study on the permeability of the breached fault seal is equivalent to examining the effect of fault core thickness on the strength of the gas kick."

*CharLotte Krawczyk (Executive Editor)*

Dear Dominik,

in addition to the corrections advised by Tarje Nissen-Meyer, there is also one reviewer comment and your answer, I would like to direct your attention to: "We would like to clarify that, although the St. Gallen deep geothermal project has been considered an EGS in some studies (e.g., Breede et al.,2013), we prefer to classify it as a hydrothermal project, as no

hydraulic stimulation for targeted shearing of fractures (i.e., hydro-shearing) adjacent to the injection well was performed." This thought may arise for several readers, so that I'd suggest to include your rephrased answer somewhere in the beginning of your final revised manuscript, so that a statement clarifies your view.

I'm looking forward seeing your manuscript published in SE, Lotte.

We thank CharLotte Krawczyk for this suggestion. We have now clarified our view in Sect. 2 as follows:

"Although the St. Gallen deep geothermal project has been considered an EGS in some studies (Breede et al., 2013), we here clearly classify it as a hydrothermal project, since no hydraulic stimulation for the targeted shearing of fractures (hydro-shearing) was performed adjacent to the injection well (see below)."